# BAG2 promotes tumorigenesis through enhancing mutant p53 protein levels and function

**Xuetian Yue[1†], Yuhan Zhao[1†], Juan Liu[1], Cen Zhang[1], Haiyang Yu[1], Jiabei Wang[1,2], Tongsen Zheng[1,2], Lianxin Liu[2], Jun Li[1], Zhaohui Feng[1]\*, Wenwei Hu[1,3]\***

[1]Rutgers Cancer Institute of New Jersey, Rutgers University, New Brunswick, United States; [2]Key Laboratory of Hepatosplenic Surgery, Harbin Medical University, Harbin, China; [3]Department of Pharmacology, Rutgers University, New Brunswick, United States

**Abstract** Tumor suppressor p53 is the most frequently mutated gene in tumors. Many mutant p53 (mutp53) proteins promote tumorigenesis through the gain-of-function (GOF) mechanism. Mutp53 proteins often accumulate to high levels in tumors, which is critical for mutp53 GOF. Its underlying mechanism is poorly understood. Here, we found that BAG2, a protein of Bcl-2 associated athanogene (BAG) family, promotes mutp53 accumulation and GOF in tumors. Mechanistically, BAG2 binds to mutp53 and translocates to the nucleus to inhibit the MDM2-mutp53 interaction, and MDM2-mediated ubiquitination and degradation of mutp53. Thus, BAG2 promotes mutp53 accumulation and GOF in tumor growth, metastasis and chemoresistance. BAG2 is frequently overexpressed in tumors. BAG2 overexpression is associated with poor prognosis in patients and mutp53 accumulation in tumors. These findings revealed a novel and important mechanism for mutp53 accumulation and GOF in tumors, and also uncovered an important role of BAG2 in tumorigenesis through promoting mutp53 accumulation and GOF.

*For correspondence: fengzh@ cinj.rutgers.edu (ZF); wh221@cinj. rutgers.edu (WH)

†These authors contributed equally to this work

Competing interests: The authors declare that no competing interests exist.

## Introduction

Tumor suppressor p53 plays a central role in tumor prevention (*Levine et al., 2006*; *Levine and Oren, 2009*; *Vousden and Prives, 2009*). Trp53 is the most frequently mutated gene in human tumors; it is mutated in over 50% of all tumors. Majority of Trp53 mutations are missense mutations that are localized in the p53 DNA binding domain (DBD), including several mutational hotspots in tumors (e.g., R175, R248, and R273) (*Harris and Hollstein, 1993*; *Freed-Pastor and Prives, 2012*; *Muller and Vousden, 2014*). Many tumor-associated mutant p53 (mutp53) proteins not only lose the tumor suppressive function of wild-type p53 (wtp53), but also gain new oncogenic activities independently of wtp53, which is defined as mutp53 gain-of-function (GOF) (*Freed-Pastor and Prives, 2012*; *Muller and Vousden, 2014*). So far, many mutp53 GOFs have been identified, including promoting tumor growth, metastasis, chemoresistance and metabolic changes (*Lang et al., 2004*; *Olive et al., 2004*; *Muller et al., 2009*; *Blandino et al., 2012*; *Freed-Pastor et al., 2012*; *Cooks et al., 2013*; *Zhang et al., 2013*).

Under the non-stressed condition, wtp53 protein levels are kept low in normal cells and tissues mainly through the proteasomal degradation mediated by E3 ubiquitin ligase MDM2, the most critical negative regulator for wtp53 (*Brooks and Gu, 2006*; *Hu et al., 2012*). At the same time, as a direct transcriptional target of p53, MDM2 is up-regulated by p53 under both non-stressed and stressed conditions. Thus, p53 and MDM2 forms a negative feedback loop to tightly regulate p53 protein levels in cells. However, mutp53 proteins often become stable and accumulate to high levels in

**eLife digest** Cancer can develop if cells in the body acquire mutations that enable them to grow rapidly to form a mass called a tumor. The gene that encodes a protein called p53 is the most commonly mutated gene in human tumors. Most of these mutations result in the production of mutant p53 proteins that are similar in size to the normal protein, but do not work in the same way.

The normal p53 protein is known as a 'tumor suppressor' because it promotes the repair of damaged genetic material and stops the cell from dividing while this repair is underway. Also, it can instruct a cell to die if the damage is too great to repair. However, many of the mutant p53 proteins stop performing these roles and gain new activities that promote tumor growth instead. These activities often rely on the mutant p53 proteins accumulating to very high levels, but it is not clear why this happens.

Here, Yue, Zhao et al. used biochemical techniques to search for other proteins that can bind to mutant p53 proteins. The experiments reveal that a protein called BAG2 binds to mutant p53 and promotes its accumulation in cancer cells, which increases the activity of mutant p53 in driving tumor growth. Loss of BAG2 leads to a reduction in the level of mutant p53 in cells and inhibits the activity of mutant p53.

Using a public database of genetic data from human tumors, Yue, Zhao et al. found that human tumor cells often contain higher levels of BAG2 than normal cells. Furthermore, patients with tumors that had high levels of BAG2—and therefore accumulated mutant p53 proteins—were less likely to have positive outcomes after medical treatment.

Yue, Zhao et al.'s findings suggest that increased production of BAG2 in many tumors may be responsible for the accumulation of mutant p53 proteins that drive tumor growth. A future goal is to develop a new treatment strategy that targets BAG2 in tumors to prevent the accumulation of mutant p53 proteins and therefore block the growth of tumors.

tumors, which is critical for mutp53 GOF in tumorigenesis and contributes greatly to tumor progression (*Oren and Rotter, 2010*; *Li et al., 2011a*; *Freed-Pastor and Prives, 2012*; *Muller and Vousden, 2013*). It had long been thought that the inability of MDM2 to degrade mutp53 was the main cause for mutp53 protein accumulation in tumors. However, recent studies from mice with knock-in of R172H or R270H mutp53 (equivalent to human R175H and R273H mutp53, respectively) challenged this concept. Mutp53 protein is kept at low levels in normal tissues but accumulates to very high levels in tumors (*Lang et al., 2004*; *Olive et al., 2004*). Furthermore, loss of MDM2 in mutp53 knock-in mice leads to mutp53 protein accumulation in normal tissues, which in turn promotes tumor development (*Terzian et al., 2008*). Recent studies including ours also showed that MDM2 retains the ability to degrade mutp53 in in vitro cultured cells (*Lukashchuk and Vousden, 2007*; *Zheng et al., 2013*). These results strongly suggest that while MDM2 maintains mutp53 protein levels low in normal tissues, the disruption of MDM2-mediated mutp53 degradation in tumors could be a main cause for the frequently observed mutp53 protein accumulation in tumors. Currently, the mechanism underlying the disruption of MDM2-mediated mutp53 degradation in tumors is poorly understood. Destabilizing mutp53 to inhibit mutp53 GOF is being actively tested as a novel and promising strategy for cancer therapy. Understanding the underlying mechanism for mutp53 accumulation is critical for the development of novel targets and strategies for cancer therapy.

In this study, to investigate the mechanism underlying mutp53 accumulation in tumors, we screened for proteins interacting with mutp53 using liquid chromatography-tandem mass spectrometry (LC-MS/MS) assays in tumors from R172H mutp53 knock-in mice, and identified BAG2 as a novel mutp53 binding protein that plays a critical role in promoting mutp53 accumulation in tumors. BAG2 belongs to the Bcl-2 associated athanogene (BAG) family, which is characterized by the BAG domain. As a group of multifunctional proteins, BAG proteins interact with a variety of proteins and take part in diverse cellular processes, including cell division, cell death and differentiation (*Takayama and Reed, 2001*; *Kabbage and Dickman, 2008*). Currently, the role of BAG2 in tumorigenesis and its underlying mechanism are poorly understood. We found that mutp53 binds to BAG2 and promotes the nuclear translocation of BAG2. The BAG2-mutp53 interaction in the nucleus inhibits the ubiquitination and degradation of mutp53 mediated by MDM2, and thereby

promotes mutp53 accumulation and mutp53 GOF in tumorigenesis. Knockdown of BAG2 greatly decreases mutp53 protein levels in tumors and compromises mutp53 GOF in tumorigenesis. BAG2 is frequently overexpressed in various types of human tumors. BAG2 overexpression is associated with poor prognosis in cancer patients and mutp53 accumulation in tumors. These results revealed a novel and critical mechanism for mutp53 protein accumulation in tumors, and strongly suggest that BAG2 is a potential target for therapy in tumors carrying mutp53. Our results also uncovered an important role of BAG2 in tumorigenesis and revealed that promoting mutp53 accumulation and GOF is a novel mechanism for BAG2 in tumorigenesis.

## Results

### BAG2 is a novel mutp53-interacting protein in Trp53$^{R172H/R172H}$ mouse tumors and human cells

R172H mutp53 knock-in (Trp53$^{R172H/R172H}$) mice mainly develop lymphomas in the spleen and thymus (*Lang et al., 2004*; *Olive et al., 2004*). Mutp53 protein levels are drastically increased in majority of tumors from Trp53$^{R172H/R172H}$ mice but are very low in normal tissues. To investigate the mechanism underlying mutp53 accumulation in tumors, we screened for proteins interacting with mutp53 in thymic lymphomas of Trp53$^{R172H/R172H}$ mice with drastic mutp53 accumulation (n = 3) using immunoprecipitation (IP) assays with an anti-p53 antibody followed by LC-MS/MS assays (*Figure 1A*). Normal tissues of Trp53$^{R172H/R172H}$ mice with low mutp53 levels were used as controls. LC-MS/MS assays identified a list of potential proteins binding to mutp53 in the thymic lymphomas of Trp53$^{R172H/R172H}$ mice (*Figure 1B*). Several known mutp53-binding proteins, including HSP90, Myosin, Cct8 and Pontin (*Muller et al., 2005*; *Trinidad et al., 2013*; *Arjonen et al., 2014*; *Zhao et al., 2015*), were among the list of proteins identified in tumors of Trp53$^{R172H/R172H}$ mice, which validated our approach. The complete list of proteins that bound to mutp53 in Trp53$^{R172H/R172H}$ tumors was listed in *Table 1*.

Interestingly, BAG2 was identified as a potential mutp53 binding protein (*Figure 1B*). The BAG2-mutp53 interaction in Trp53$^{R172H/R172H}$ tumors was confirmed by co-IP followed by Western blot assays (*Figure 1C*). The weak interaction between BAG2 and mutp53 was also observed in normal tissues from Trp53$^{R172H/R172H}$ mice, including thymus, spleen and kidney (*Figure 1C*, *Figure 1—figure supplement 1*). To investigate whether BAG2 specifically interacts with mutp53 in human cells, human p53-null lung cancer H1299 cells were transfected with human BAG2-HA expression vectors together with human wtp53 or mutp53 (R175H) expression vectors. Co-IP assays employing either anti-p53 or anti-HA antibodies showed that BAG2 preferentially bound to mutp53 compared with wtp53 (*Figure 2A*). In addition to R175H, the strong BAG2-mutp53 interaction was observed in H1299 cells with ectopic expression of different mutp53 proteins, including R248W and R273H, respectively (*Figure 2B*). The interaction between the endogenous BAG2 and mutp53 proteins was also observed in several human cancer cell lines, including human colorectal cancer HCT116 p53$^{R248W/-}$, HT-29 and SW480 cell lines which contain a single copy of Trp53 gene with R248W and R273H mutation, respectively, human breast cancer SK-BR-3, MDA-MB-468 cell lines which contain a single copy of Trp53 gene with R175H and R273H mutation, respectively, and human hepatocellular carcinoma Huh-7 cell lines which contain a single copy of Trp53 gene with Y220C mutation (*Figure 2C*, *Figure 2—figure supplement 1*). Together, these results demonstrate that BAG2 is a novel mutp53-specific binding partner, and this interaction is conserved in both mouse tumors and human cancer cells.

### DBD of mutp53 and BAG domain of BAG2 are essential for the BAG2-mutp53 interaction

p53 protein contains two transcriptional activation domains (AD1 and AD2), a sequence-specific DBD, a tetramerization domain and a C-terminal domain (C-ter). To define the regions of mutp53 required for the BAG2-mutp53 interaction, expression vectors of fragments containing different mutp53 domains with HA-tag (*Figure 2D*, upper panel) and BAG2-Flag expression vectors were co-transfected into p53-null H1299 cells. Results of co-IP assays using an anti-Flag antibody showed that BAG2 interacted with all mutp53 (R175H) fragments containing the mutp53 DBD (P1-P5 in *Figure 2D*), but not the fragment lacking the mutp53 DBD (P6 in *Figure 2D*). Furthermore, BAG2

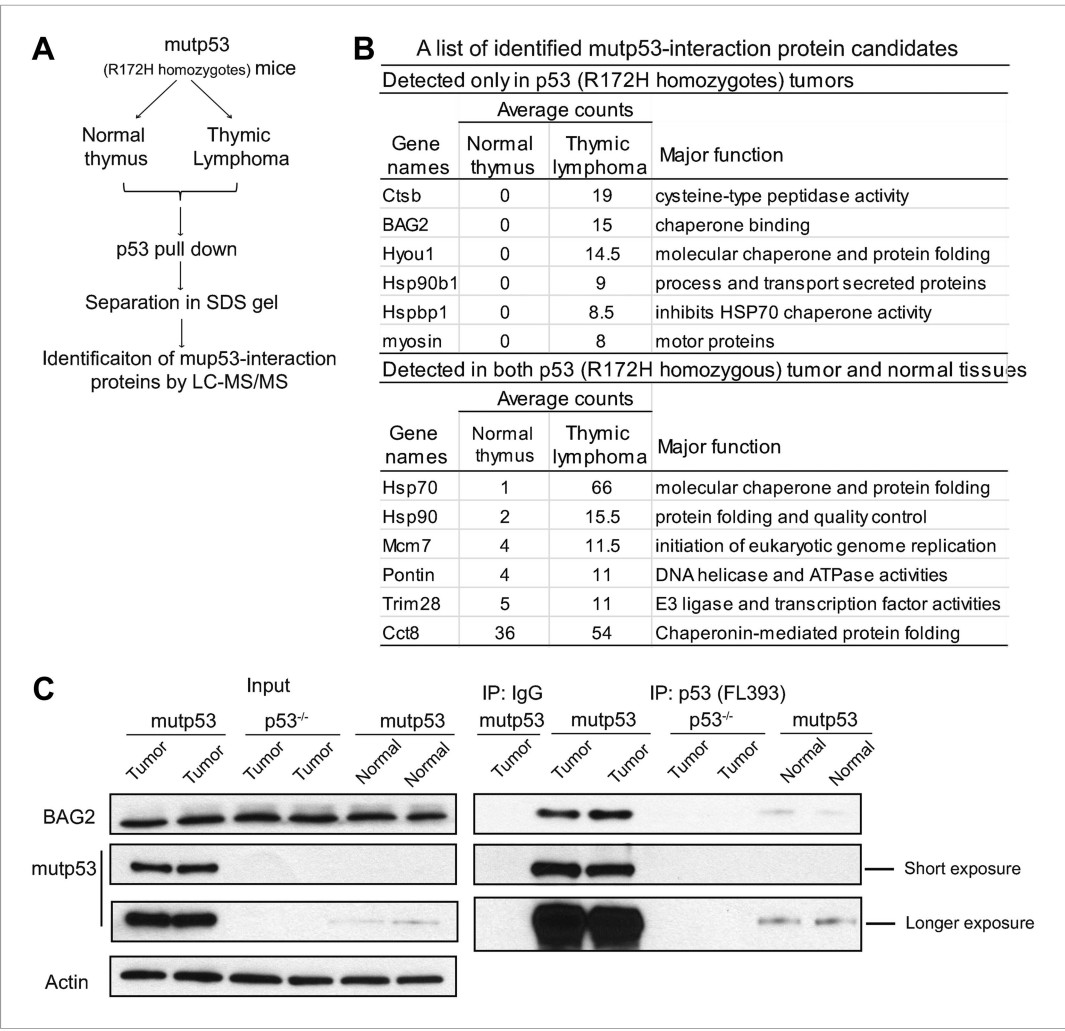

Figure 1. Identification of proteins interacting with mutant p53 (mutp53) protein in tumors from Trp53$^{R172H/R172H}$ mice. (A) Work flow for identification of proteins interacting with mutp53 protein. Lysate of thymic lymphomas and normal thymus from Trp53$^{R172H/R172H}$ mice were subjected to co-immunoprecipitation (co-IP) using anti-p53 (FL393) beads. Eluted proteins were separated in a 4–15% SDS PAGE gel and analyzed by LC-MS/MS. (B) The table of a list of protein candidates that interacted with mutp53 protein. (C) The interaction of mutp53 with BAG2 in thymic lymphomas of Trp53$^{R172H/R172H}$ mice was confirmed by co-IP assays followed by Western blot assays. Thymic lymphomas from Trp53$^{R172H/R172H}$ mice and p53$^{-/-}$ mice as well as normal thymic tissue from Trp53$^{R172H/R172H}$ mice were subjected to co-IP assays using an anti-p53 antibody.

The following figure supplement is available for figure 1:

Figure supplement 1. The interaction of mutp53 with BAG2 in normal mouse tissues of Trp53$^{R172H/R172H}$ mice.

preferentially bound to DBDs of different mutp53 (R175H, R248W and R273H) but not wtp53 DBD (Figure 2E, Figure 2—figure supplement 2).

The regions of BAG2 required for the BAG2-mutp53 interaction was examined by co-transfecting cells with vectors expressing different Flag-tagged BAG2 deletion mutants (Figure 2F, left panel) and mutp53 (R175H) expression vectors followed by co-IP assays. BAG2 contains a BAG domain (amino acids 91–211) at the C-terminus (Dai et al., 2005). The fragments containing the BAG domain interacted with mutp53 while the N-terminus of BAG2 protein lacking the BAG domain did not interact with mutp53 (Figure 2F). Interestingly, the binding of mutp53 to the BAG2 fragment which lacks the N-terminus is much weaker compared with the full length (FL) BAG2 protein. It is possible

**Table 1.** The list of identified mutp53-interaction protein candidates

| Gene names | Average counts | |
|---|---|---|
| | Normal thymus | Thymic lymphoma |
| Ctsb | 0 | 19 |
| Tfrc | 0 | 15.5 |
| Bag2 | 0 | 15 |
| Stip1 | 0 | 14.5 |
| Hyou1 | 0 | 14.5 |
| Cad | 0 | 14.5 |
| Rps19 | 0 | 14.5 |
| Pfn1 | 0 | 14 |
| Cand1 | 0 | 11.5 |
| Hspa2 | 0 | 11 |
| Lcp1 | 0 | 11 |
| Sar1a | 0 | 10.5 |
| Fam49b | 0 | 10.5 |
| Khsrp | 0 | 10 |
| Ifi47 | 0 | 9.5 |
| Cse1l | 0 | 9.5 |
| Ipo5 | 0 | 9.5 |
| Hsp90b1 | 0 | 9 |
| Hspbp1 | 0 | 8.5 |
| Rfc5 | 0 | 8.5 |
| Tkt | 0 | 8.5 |
| myosin | 0 | 8 |
| Hadhb | 0 | 8 |
| Hsp70 | 1 | 66 |
| Phgdh | 1 | 20 |
| Myh9 | 5 | 81.5 |
| Hspd1 | 2 | 32.5 |
| Rpl9-ps4 | 1 | 16 |
| Ubr5 | 1 | 13.5 |
| Dars | 1 | 13.5 |
| Iqgap1 | 1 | 12 |
| Slc25a3 | 1 | 11.5 |
| Rars | 1 | 11.5 |
| Ruvbl2 | 1 | 11 |
| Ddb1 | 1 | 10 |
| Hsph1 | 4 | 38.5 |
| Dnajb4 | 1 | 9 |
| Aldoa | 1 | 8.5 |
| Pcna | 2 | 16.5 |
| Eprs | 1 | 8 |
| Hsp90 | 2 | 15.5 |

*Table 1. Continued on next page*

that the N-terminus of BAG2 has an additional role for efficient BAG2-mutp53 complex formation although itself does not directly interact with mutp53. These results demonstrate that mutp53 DBD and BAG domain of BAG2 are essential for the BAG2-mutp53 interaction.

## BAG2 promotes mutp53 protein accumulation in cancer cells

It was reported that BAG2 stabilizes some of its binding proteins, such as PINK1 and ataxin3-80Q (*Che et al., 2013*, *2014*). To investigate whether BAG2 regulates mutp53 protein levels, endogenous BAG2 was knocked down by 2 different siRNA oligos and its impact upon mutp53 protein levels was evaluated in HCT116 p53[R248W/−] cells and p53-null Saos2 cells with stable ectopic expression of different mutp53 (Saos2-R175H, Saos2-R248W and Saos2-R273H). The knockdown of BAG2 was confirmed at both mRNA and protein levels by real-time PCR and Western blot assays, respectively (*Figure 3A,B*). While BAG2 knockdown showed no apparent effect on mutp53 mRNA levels (*Figure 3—figure supplement 1*), BAG2 knockdown greatly decreased the mutp53 protein levels in cells (*Figure 3A*). The effect of BAG2 over-expression on mutp53 protein levels was also determined in these cells. Ectopic BAG2 expression by vectors clearly increased mutp53 protein levels (*Figure 3C*), while had no clear effect on mutp53 mRNA levels in cells (*Figure 3—figure supplement 2*). These results demonstrate that BAG2 increases mutp53 protein levels in cells.

## BAG2 inhibits the degradation of mutp53 protein mediated by MDM2

BAG2 is a co-chaperone protein, which can regulate the ubiquitination and degradation of some proteins (*Dai et al., 2005*; *Che et al., 2013*). Here, we investigated whether BAG2 promotes mutp53 protein accumulation through the inhibition of mutp53 protein ubiquitination and degradation. Since endogenous BAG2 expression levels are relatively higher in Saos2 and HCT116 p53[R248W/−] cells compared with H1299 cells as determined at the RNA and protein levels (*Figure 3—figure supplement 3*), experiments with knockdown of endogenous BAG2 were performed by using Saos2 and HCT116 p53[R248W/−] cells, and experiments with ectopic BAG2 expression were performed by using H1299 cells. We found that blocking proteasomal degradation by the proteasome inhibitor MG132 largely abolished the effect of BAG2 knockdown on mutp53 protein levels in HCT116 p53[R248W/−], Saos2-R175H, Saos2-R248W and Saos2-R273H cells

Table 1. Continued

| Gene names | Average counts | |
| --- | --- | --- |
| | Normal thymus | Thymic lymphoma |
| Gm9755 | 2 | 12.5 |
| Dnaja1 | 3 | 18 |
| Atp5b | 3 | 18 |
| Cltc | 7 | 41 |
| Gm5506 | 5 | 26 |
| Dnaja2 | 3 | 15 |
| Bag5 | 7 | 31.5 |
| Rps7 | 5 | 22.5 |
| Ywhae | 2 | 9 |
| Eef2 | 10 | 38.5 |
| Adsl | 2 | 7.5 |
| Hsp90ab1 | 20 | 74.5 |
| Gnb2l1 | 6 | 22 |
| Copg | 2 | 7 |
| Rpl23 | 2 | 7 |
| Psmc6 | 2 | 7 |
| Pcbp2 | 2 | 7 |
| Pcbp1 | 3 | 10 |
| Pabpc4 | 5 | 15.5 |
| Hspa8 | 77 | 237.5 |
| Fcgr4 | 13 | 38.5 |
| Mcm7 | 4 | 11.5 |
| Hadha | 3 | 8.5 |
| Kpnb1 | 5 | 14 |
| Atp5a1 | 9 | 25 |
| Pontin | 4 | 11 |
| Bat3 | 3 | 8 |
| Pdia6 | 3 | 7.5 |
| Dnajc7 | 38 | 86 |
| Rps15a | 6 | 13.5 |
| Aldh2 | 4 | 9 |
| Trim28 | 5 | 11 |
| Eef1a1 | 16 | 35 |
| St13 | 6 | 13 |
| Cct8 | 36 | 54 |
| Psmd11 | 6 | 9 |

(*Figure 3D*). Ectopic expression of MDM2 clearly down-regulated mutp53 R175H in H1299 cells co-transfected with vectors expressing mutp53 R175H and MDM2 (*Figure 3E*), which is consistent with previous reports (*Lukashchuk and Vousden, 2007*; *Zheng et al., 2013*). Notably, co-expression of BAG2 largely reduced the degradation of mutp53 protein mediated by MDM2 (*Figure 3E*). Consistently, knockdown of endogenous MDM2 clearly increased mutp53 protein levels in Saoa2-R175H cells (*Figure 3F*). Notably, the effect of BAG2 knockdown on mutp53 protein levels was greatly reduced in cells with MDM2 knockdown, indicating that the effect of BAG2 knockdown on mutp53 protein levels is largely mediated by MDM2 (*Figure 3F*). MDM2 directly binds to mutp53 to negatively regulate mutp53. Co-expression of BAG2 clearly decreased the interaction of MDM2 with mutp53 in H1299 cells, which could be an important mechanism by which BAG2 inhibits MDM2-mediated mutp53 degradation (*Figure 3G*).

To investigate whether BAG2 regulates mutp53 protein through inhibiting mutp53 ubiquitination, in vivo ubiquitination assays were employed. Ectopic BAG2 expression reduced ubiquitination of mutp53 in H1299 cells (*Figure 3H*). Knockdown of endogenous BAG2 by siRNA increased ubiquitination of mutp53 in Saos2-R175H cells (*Figure 3I*). These results demonstrate that BAG2 interacts with mutp53, and inhibits MDM2 binding to and degradation of mutp53, which leads to the mutp53 accumulation in cells.

## Mutp53 promotes the nuclear translocation of BAG2

It has been reported that BAG2 proteins were mainly localized in the cytoplasm (*Dai et al., 2005*). Indeed, in H1299 cells with ectopic expression of BAG2 alone, BAG2 proteins were predominantly localized in the cytoplasm as determined by immunofluorescence (IF) staining (*Figure 4A*). Interestingly, we found that mutp53 promoted BAG2 nuclear translocation; ectopic expression of mutp53 (R175H, R248W and R273H), which is mainly localized in the nucleus, clearly increased the translocation of BAG2 from the cytoplasm to the nucleus in cells transfected with vectors expressing BAG2 together with mutp53. Furthermore, BAG2 was largely co-localized with mutp53 in the nucleus (*Figure 4A*). In contrast, ectopic expression of wtp53, which is also mainly localized in the nucleus, did not have an obvious effect on BAG2 nuclear translocation in cells (*Figure 4A*). The effect of mutp53 on BAG2 nuclear translocation was also confirmed by Western blot assays using whole cell lysates and nuclear extracts isolated from H1299 cells transfected with BAG2 vectors

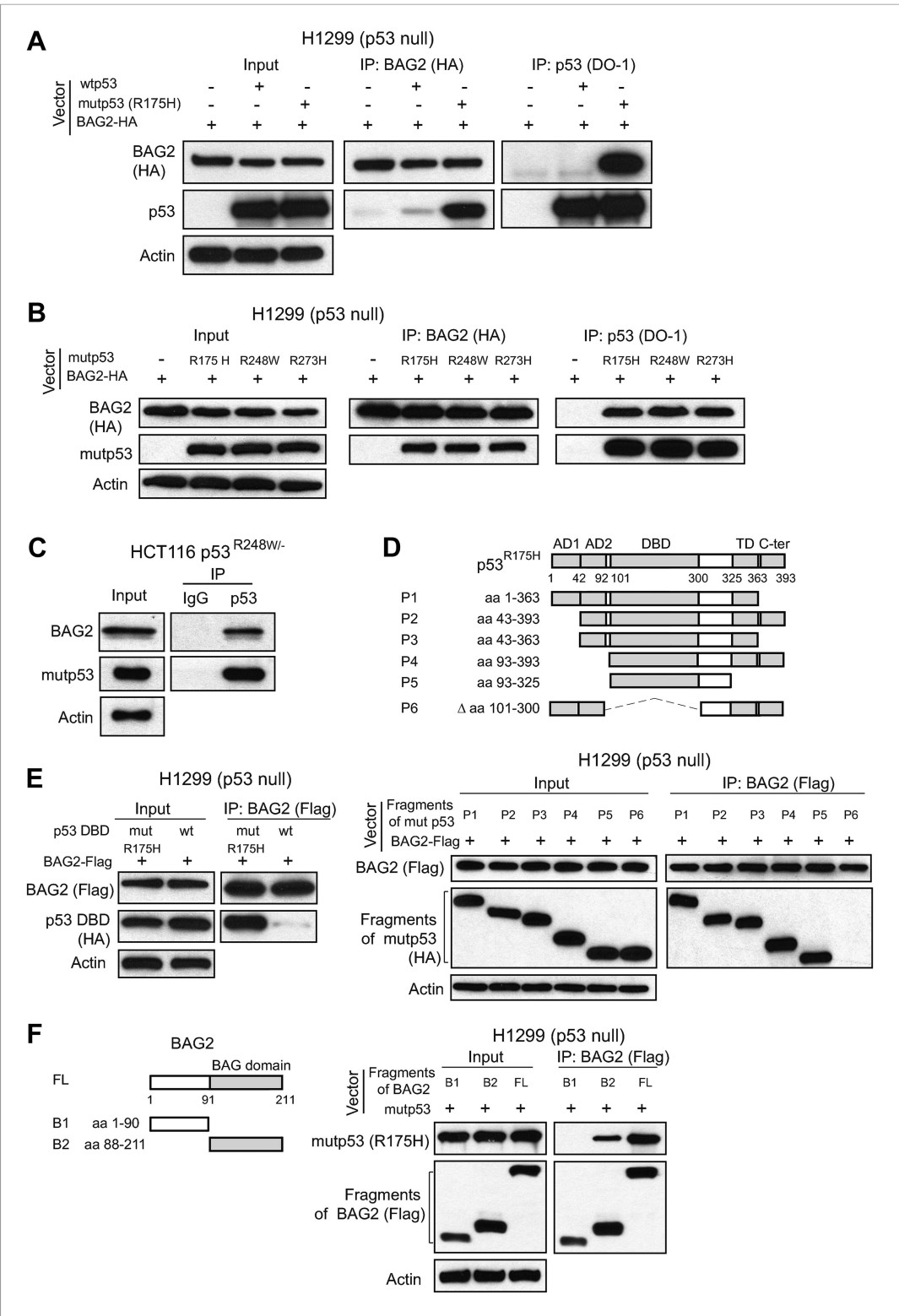

**Figure 2**. BAG2 is a mutp53-specific binding partner as determined by reciprocal co-IP assays in human cell lines. (**A**) Ectopically expressed BAG2 preferentially interacted with mutp53 (R175H) protein compared with wild-type p53 (wtp53) protein in H1299 cells. H1299 cells were transiently transfected with vectors expressing mutp53 (R175H) or wtp53 together with HA-tagged BAG2 (BAG2-HA) expression vectors. Antibodies used for IP assays: HA for BAG2-HA and DO-1 for p53. (**B**) BAG2 interacted with several hotspot mutp53 proteins (R175H, R248W and R273H) in

*Figure 2. continued on next page*

*Figure 2. Continued*

H1299 cells. H1299 cells were transiently transfected with vectors expressing mutp53 (R175H, R248W or R273H) together with BAG2-HA expression vectors. (**C**) The interaction of endogenous BAG2 with mutp53 (R248W) was observed in human colorectal cancer HCT116 p53$^{R248W/-}$ cells containing one allele of mutant *p53* gene (R248W). (**D**) BAG2 interacted with mutp53 DNA binding domain (DBD). Upper panel: Schematic diagram showing the domain structure of mutp53 (R175H). Lower Panel: H1299 cells were transiently transfected with expression vectors of HA-tagged mutp53 (R175H) fragments together with BAG2-Flag expression vectors. Antibodies used for IP: Flag for BAG2-Flag proteins. (**E**) BAG2 preferentially interacted with the DBD of mutp53 (R175H) but not wtp53 DBD. H1299 cells were transiently transfected with expression vectors of HA-tagged mutp53 (R175H) DBD or wtp53 DBD together with BAG2-Flag expression vectors. (**F**) Mutp53 interacted with the Bcl-2 associated athanogene (BAG) domain of BAG2. Left panel: Schematic diagram showing the domain structure of BAG2. Right panel: H1299 cells were transiently transfected with expression vectors of mutp53 (R175H) together with Flag-tagged BAG2 fragments.

The following figure supplements are available for figure 2:

**Figure supplement 1**. The interaction of endogenous BAG2 with mutp53 in several human tumor cell lines containing endogenous mutp53.

**Figure supplement 2**. The interaction of BAG2 with mutp53 (R248W and R273H) DBD in H1299 cells.

---

alone or together with mutp53 vectors (*Figure 4B*). Both mutp53 and MDM2 proteins contain a nuclear localization signal (NLS) and are mainly localized in the nucleus, where MDM2 binds to and ubiquitinates mutp53 protein. The translocation of BAG2 to the nucleus where it interacts with mutp53 may play an important role in blocking MDM2 to bind to and degrade mutp53. To test this possibility, we constructed the vector expressing the NLS mutant of mutp53 R175H (mutp53$^{NLS}$) by mutating Lys305, Arg306, Lys319, Lys320 and Lys321 to Ala as reported (*O'Keefe et al., 2003*). Unlike mutp53 proteins which were mainly localized in the nucleus, mutp53$^{NLS}$ proteins were mainly localized in the cytoplasm as shown by IF staining (*Figure 4A*). While mutp53$^{NLS}$ readily interacted with BAG2 as determined by co-IP assays (*Figure 4C*), mutp53$^{NLS}$ could not promote the nuclear translocation of BAG2. BAG2 was mainly localized in the cytoplasm in H1299 co-transfected with vectors expressing BAG2 and mutp53$^{NLS}$ (*Figure 4A*). Notably, ectopic expression of MDM2 showed a limited effect on degradation of mutp53$^{NLS}$ protein compared with mutp53 (R175H) (*Figure 4D*). Furthermore, co-expression of BAG2 had no obvious effect on mutp53$^{NLS}$ protein levels in cells co-transfected with expression vectors of BAG2, mutp53$^{NLS}$ and MDM2 (*Figure 4D*). These results strongly suggest that mutp53 promotes BAG2 nuclear localization and the BAG2-mutp53 interaction in the nucleus inhibits MDM2-mediated mutp53 protein degradation.

## BAG2 promotes mutp53 GOF in chemoresistance

The accumulation of mutp53 proteins is critical for mutp53 GOF in tumorigenesis (*Blandino et al., 2012*; *Muller and Vousden, 2014*). Chemoresistance is one of the most important mutp53 GOFs (*Napoli et al., 2012*; *Masciarelli et al., 2014*). 5-flurorouracil (5-FU), which can induce apoptosis in cells, is one of the most commonly used chemotherapeutic agents for a wide variety of human cancers. 5-FU induced less apoptosis in Saos2-R175H, Saos2-R248W and Saos2-R273H cells compared with Saos2-Con cells as determined by Annexin V staining and the levels of cleaved Caspase 3 protein, demonstrating that mutp53 promotes chemoresistance, which is consistent with previous reports (*Napoli et al., 2012*; *Masciarelli et al., 2014*) (*Figure 5A,B*). Notably, knockdown of BAG2 increased 5-FU-induced apoptosis in Saos2-R175H, Saos2-R248W and Saos2-R273H cells but showed a very limited effect in Saos2-Con cells (*Figure 5A,B*). Consistently, 5-FU induced less apoptosis in HCT116 p53$^{R248W/-}$ cells compared with HCT116 p53$^{-/-}$ cells. Knockdown of BAG2 increased 5-FU-induced apoptosis in HCT116 p53$^{R248W/-}$ but not HCT116 p53$^{-/-}$ cells (*Figure 5C,D*). These results demonstrate that BAG2, which promotes mutp53 protein accumulation, promotes mutp53 GOF in chemoresistance.

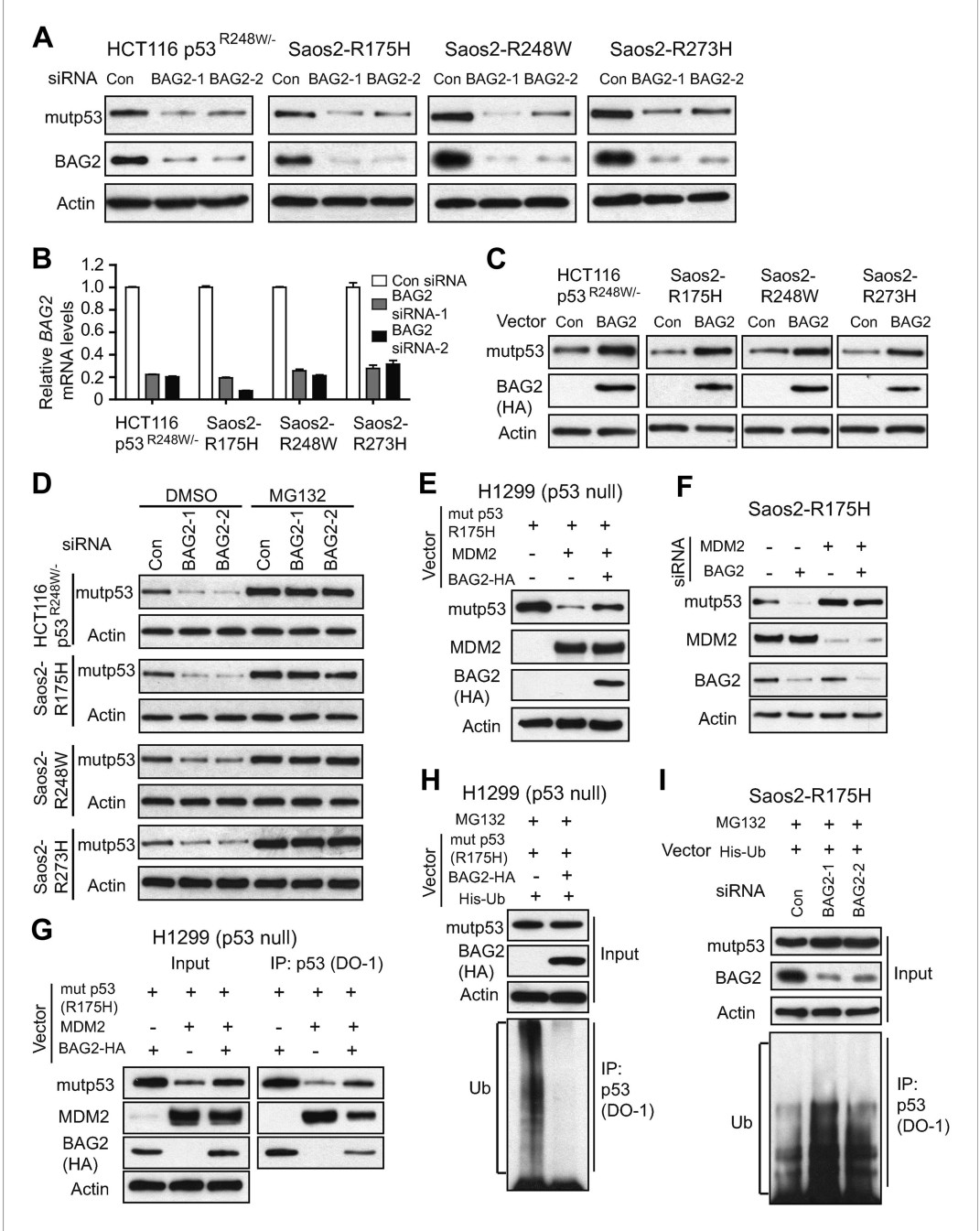

**Figure 3**. BAG2 promotes mutp53 protein accumulation in human cancer cells through the inhibition of the ubiquitination and degradation of mutp53 mediated by MDM2. (**A**) Knockdown of endogenous BAG2 by 2 different siRNA oligos decreased the mutp53 protein levels in HCT116 p53[R248W/−] and Saos2 cells with stable ectopic expression of mutp53 (Saos2-R175H, Saos2-R248W and Saos2-R273H). The knockdown of BAG2 by siRNA at the protein level was examined by Western blot assays. (**B**) The efficient knockdown of BAG2 by siRNA was confirmed at the mRNA level by real-time PCR. Data are present as mean ±SD (n = 3). (**C**) Ectopic expression of BAG2 by transfection of BAG2-HA expression vectors increased the mutp53 protein levels in cells. (**D**) Knockdown of endogenous BAG2 by siRNA decreased the mutp53 protein levels in HCT116p53[R248W/−], Saos2-R175H, Saos2-R248W and Saos2-R273H cells but not in these cells treated with the proteasome inhibitor MG132 (40 µM for 6 hr). (**E**) BAG2 inhibited the degradation of mutp53 (R175H) mediated by MDM2 in H1299 cells. Indicated combination of expression vectors of BAG2-HA, mutp53 (R175H), MDM2 were transfected into the cells. (**F**) Knockdown of MDM2 abolished the effect of BAG2 knockdown on mutp53 protein level. Knockdown of endogenous BAG2 decreased

*Figure 3. continued on next page*

*Figure 3. Continued*

mutp53 protein levels in Saos2-R175H cells but not in cells with knockdown of endogenous MDM2. (**G**) BAG2 reduced the interaction of mutp53 with MDM2 in H1299 cells as determined by IP assays. Indicated combination of expression vectors of BAG2-HA, mutp53 (R175H) and MDM2 were transfected into the cells. Antibodies used for IP: DO-1 for p53. (**H**) Ectopic BAG2 expression decreased the ubiquitination levels of mutp53 in H1299 cells. Cells were transfected with indicated combination of expression vectors of BAG2-HA, mutp53 (R175H), His-ubiquitin (His-Ub), followed by MG132 treatment. Mutp53 ubiquitination was determined by IP using DO-1 antibody (for mutp53) followed by Western blot assays using an anti-Ub antibody. (**I**) Knockdown of endogenous BAG2 increased the ubiquitination levels of mutp53 in Saos2-R175H cells. Cells were transfected with indicated combination of BAG2 siRNAs and expression vectors of His-Ub followed by MG132 treatment.

The following figure supplements are available for figure 3:

**Figure supplement 1**. Knockdown of BAG2 has no apparent effect on mutp53 mRNA expression levels in human cancer cells.

**Figure supplement 2**. Ectopic expression of BAG2 has no apparent effect on mutp53 mRNA expression levels in human cancer cells.

**Figure supplement 3**. The expression levels of BAG2 in H1299, Saos2 and HCT116 p53$^{-/-}$ cells.

## BAG2 promotes mutp53 GOF in metastasis and tumor growth

A critical GOF of mutp53 is to promote metastasis (*Lang et al., 2004*; *Olive et al., 2004*). We found that BAG2 promotes mutp53 GOF in metastasis. Migration is a critical step of metastasis. Compared with p53-null cells (Saos2-Con and HCT116 p53$^{-/-}$ cells), mutp53 (R175H, R248W and R273H in Saos2 cells and R248W in HCT116 p53$^{R248W/-}$ cells) promoted migration of cells as determined by transwell assays (*Figure 6A,B*). Notably, knockdown of BAG2 by either siRNA oligos or shRNA vectors largely abolished the promoting effect of mutp53 on migration in these cells (*Figure 6A,B*, *Figure 6—figure supplement 1*). The effect of BAG2 on mutp53 GOF in metastasis was further examined in vivo. HCT116 p53$^{R248W/-}$ and HCT116 p53$^{-/-}$ cells stably transduced with shRNA vectors against BAG2 and control cells transduced with control shRNA vectors were injected into the tail vein of nude mice to evaluate the formation of lung metastatic tumors. Mutp53 (R248W) greatly promoted lung metastatic tumor formation in nude mice; HCT116 p53$^{R248W/-}$ cells formed significantly higher number and larger size of tumors compared with HCT116 p53$^{-/-}$ cells (*Figure 6C*). Notably, this effect was greatly abolished by knockdown of BAG2 (*Figure 6C*). These results demonstrate that BAG2 promotes mutp53 GOF in metastasis.

The mutp53 GOFs also include the abilities to promote proliferation of tumor cells and anchorage-independent cell growth (*Zhang et al., 2013*). As shown in *Figure 6D* and *Figure 6—figure supplement 2*, mutp53 (R248W) promoted proliferation and anchorage-independent growth of HCT116 cells. Notably, knockdown of BAG2 clearly inhibited the rates of cell proliferation and anchorage-independent growth in HCT116 p53$^{R248W/-}$ but not HCT116 p53$^{-/-}$ cells. The xenograft tumorigenesis assays were further performed to investigate whether BAG2 knockdown reduced mutp53 GOF in promoting tumor growth in vivo. As shown in *Figure 6E*, knockdown of BAG2 in HCT116 p53$^{R248W/-}$ cells significantly inhibited the growth of xenograft tumors, whereas knockdown of BAG2 in HCT116 p53$^{-/-}$ had much less effect on the growth of xenograft tumors. Furthermore, knockdown of endogenous BAG2 clearly decreased mutp53 protein levels in HCT116 p53$^{R248W/-}$ tumors as determined by Western blot assays (*Figure 6F*), which is consistent with the results obtained from in vitro cultured cells. These results demonstrate that BAG2 promotes mutp53 GOFs in tumor cell growth.

## BAG2 is overexpressed in human tumors and high levels of BAG2 are associated with poor prognosis in cancer patients and mutp53 protein accumulation in human tumors

Results from our study have demonstrated that BAG2 interacts with mutp53 and inhibits mutp53 degradation, which in turn promotes mutp53 protein accumulation and enhances mutp53 GOF in

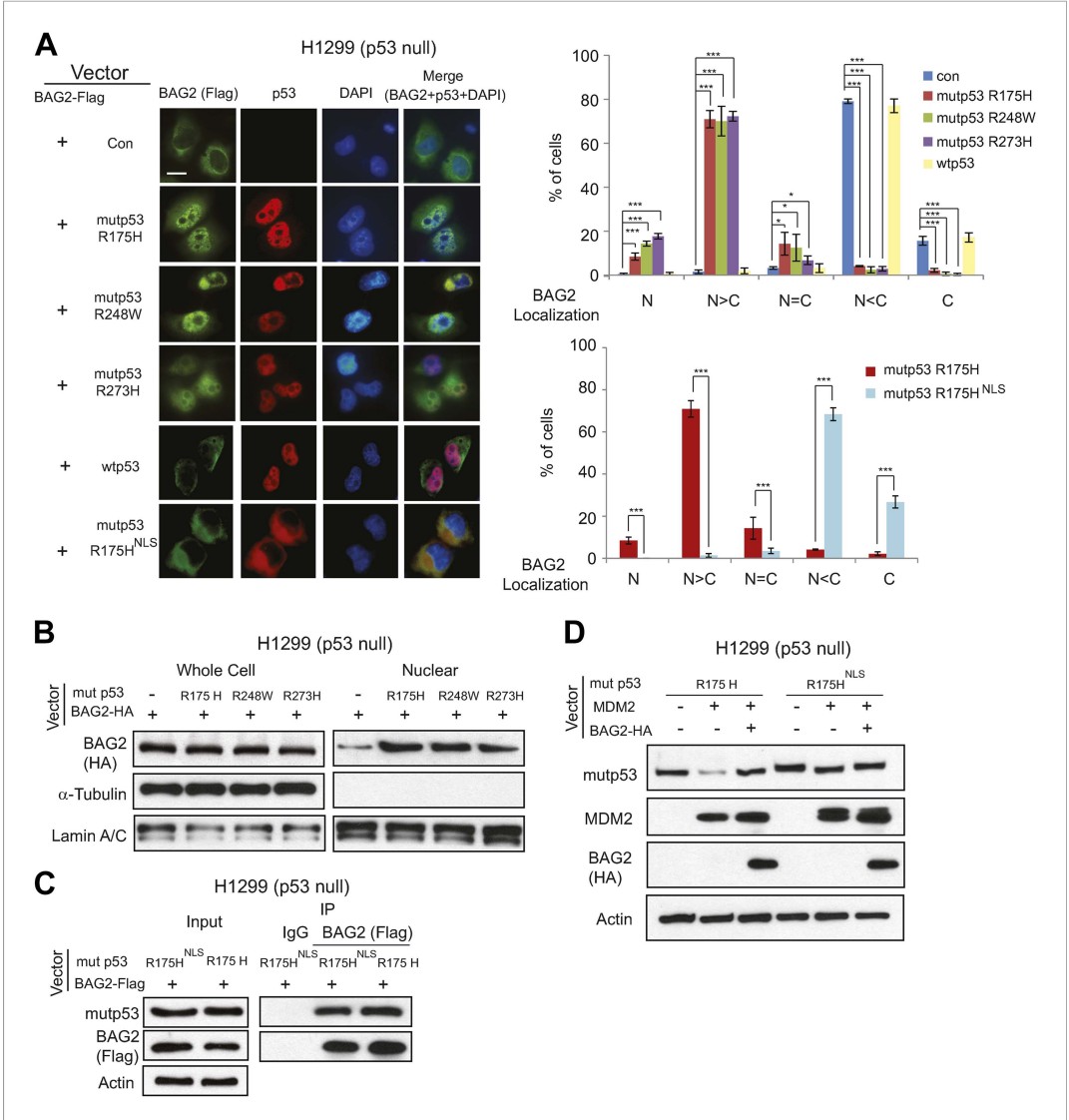

**Figure 4**. Mutp53 promotes the nuclear translocation of BAG2. (**A**) H1299 cells were transiently transfected with vectors expressing BAG2-HA together with or without expression vectors of mutp53 (R175H, R248W, R273H, or R175H[NLS]) and wtp53. The protein localization of BAG2 and p53 in cells was determined by immunofluorescence (IF) staining. Antibody used for IF: Flag for BAG2-Flag and FL393 for p53. Nuclei were stained with DAPI. Left panels: representative IF images. Scale bar: 10 μm. Right panels: quantification of the subcellular distribution of BAG2 in 200 cells for each independent experiment. Numerical data are presented in *Figure 4—source data 1*. Data are present as mean ±SD (n = 4). *p < 0.05; ***p < 0.001. (**B**) Mutp53 promotes the nuclear translocation of BAG2 in H1299 cells as determined by Western blot assays. The protein levels of BAG2 were determined in whole cell lysates and nuclear extracts prepared from H1299 cells transfected with vectors expressing BAG2-HA together with or without mutp53 (R175H, R248W or R273H). (**C**) BAG2 interacted with mutp53[NLS] (R175H[NLS]) as determined by co-IP assays. H1299 cells were transfected with vectors expressing BAG2-Flag and mutp53 R175H or mutp53 R175H[NLS]. (**D**) MDM2 had a much reduced effect on degradation of mutp53[NLS] compared with mutp53 (R175H). While BAG2 inhibited the degradation of mutp53 (R175H) mediated by MDM2, it had no obvious effect on mutp53[NLS] protein levels in H1299 cells transfected with vectors expressing BAG2-HA, MDM2 and mutp53[NLS].

The following source data is available for figure 4:

**Source data 1**. % of cells with different BAG2 localization in H1299 cells.

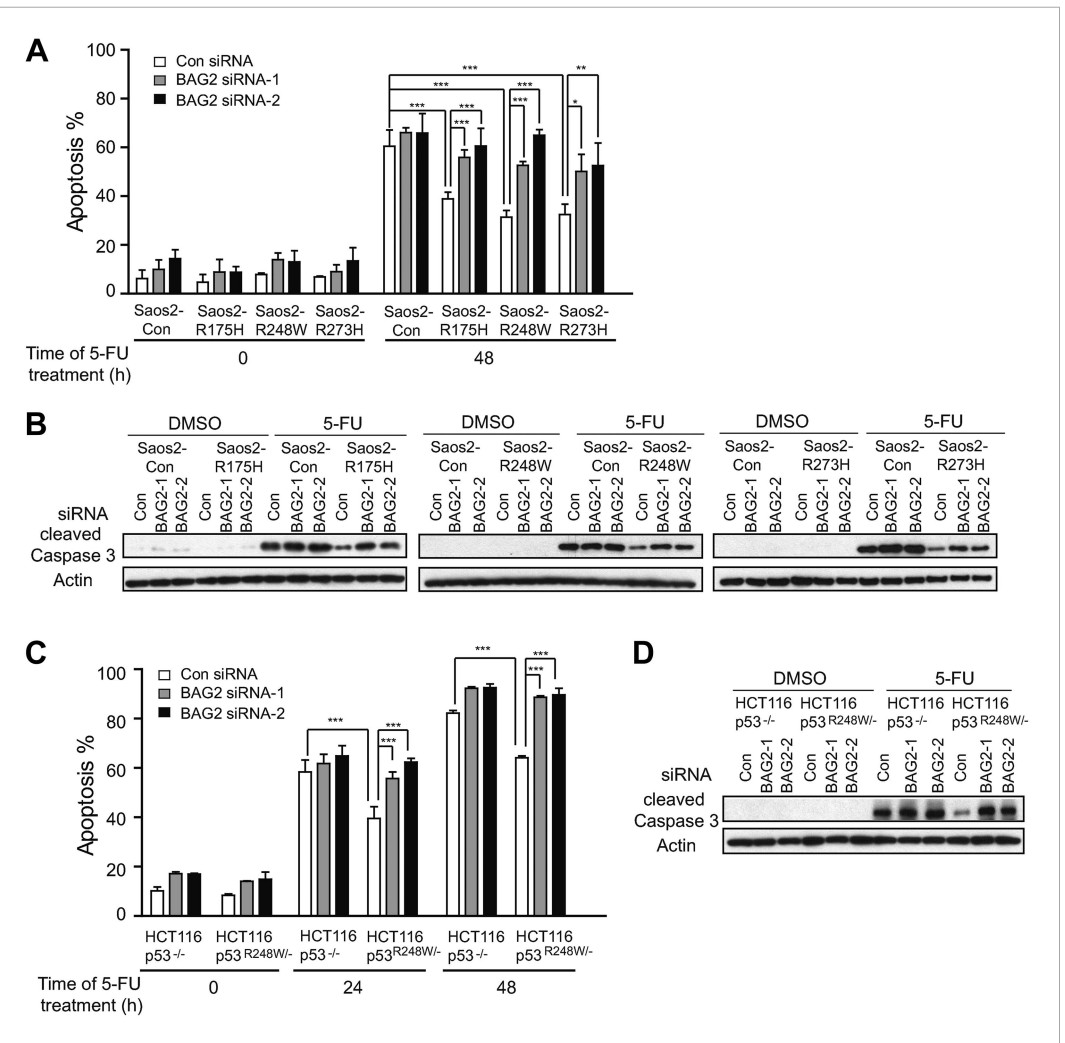

**Figure 5**. BAG2 promotes mutp53 gain-of-function (GOF) in chemoresistance. (**A**, **B**) BAG2 knockdown increased 5-FU-induced apoptosis in Saos2 cells in a largely mutp53-dependent manner. The endogenous BAG2 was knocked down by siRNA in Saos2-Con, Saos2-R175H, Saos2-R248W and Saos2-R273H cells followed by 5-FU treatment (4 mM) for 48 hr. In **A**, Annexin V assays were used to determine the percentage of apoptotic cells. Data are present as mean ±SD, n = 4. *p < 0.05; **p < 0.01; ***p < 0.001. In **B**, the levels of cleaved Caspase 3, which reflect the degree of apoptosis of cells, were determined by Western blot assays. (**C**, **D**) BAG2 knockdown increased 5-FU-induced apoptosis in HCT116 p53$^{R248W/−}$ cells but had a limited effect in HCT116 p53$^{−/−}$ cells as determined by Annexin V assays (**C**) and Western blot assays for the cleaved Caspase 3 protein levels (**D**). Numerical data for **A** and **C** are presented in *Figure 5—source data 1, 2*, respectively.

The following source data are available for figure 5:

**Source data 1**. % of apoptosis induced by 5-FU in Saos2 cells with and without ectopic expression of mutp53.

**Source data 2**. % of apoptosis induced by 5-FU in HCT116 cells with and without mutp53.

tumorigenesis. BAG2 expression was found elevated in many types of human tumors, including colorectal cancers, lung cancers, breast cancers and sarcomas, compared with normal tissues as analyzed in 4 databases from Oncomine (GSE20842, *Gaedcke et al., 2011*; GSE10072, *Landi et al., 2008*; GSE3744, *Richardson, 2006*; GSE21122, *Taylor et al., 2010*) (*Figure 7A*). The amplification of BAG2 was observed in many types of human tumors as analyzed by employing the cBioportal for Cancer Genomics (*Figure 7—figure supplement 1*), suggesting that gene amplification is an

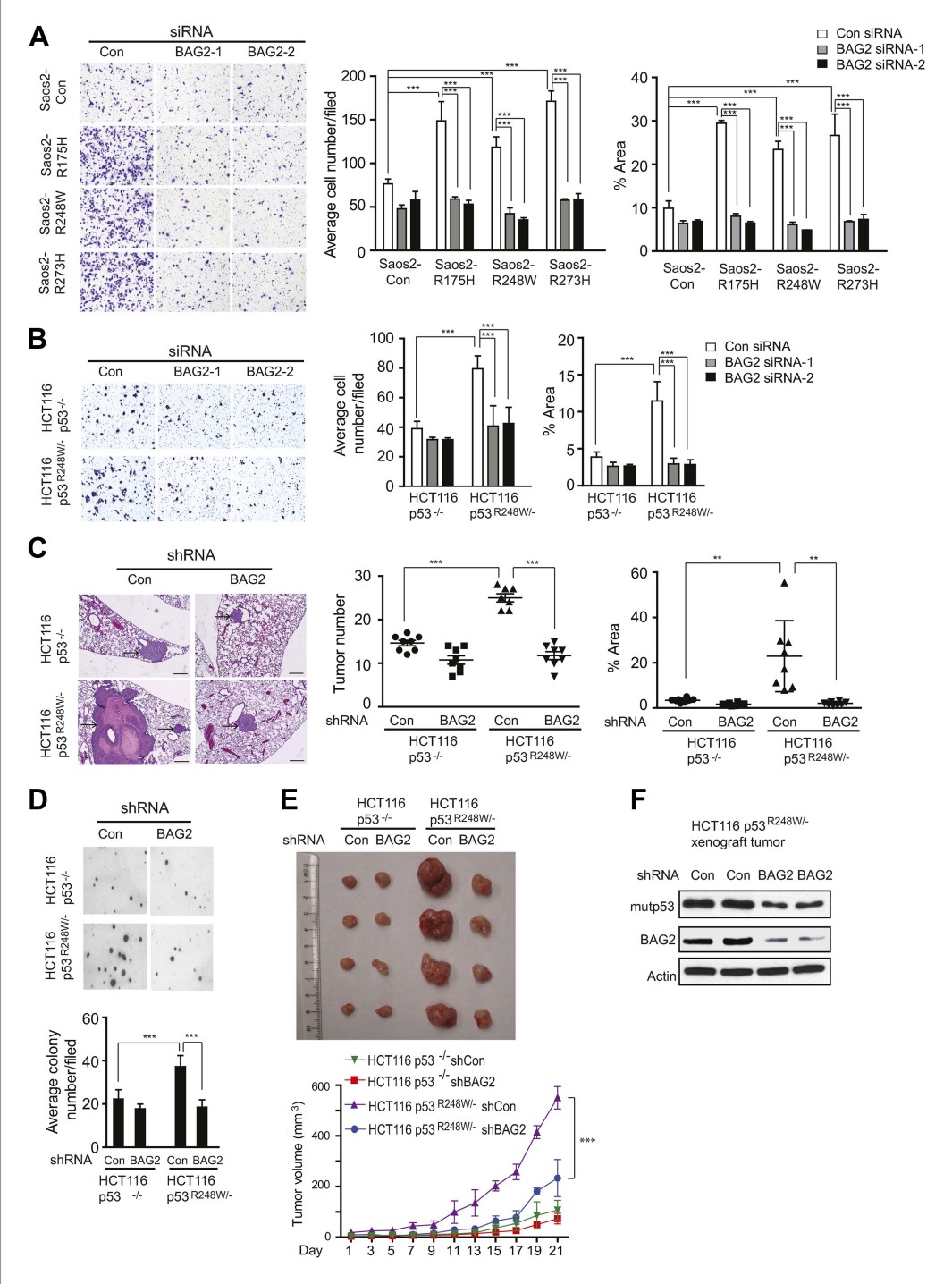

**Figure 6**. BAG2 promotes mutp53 GOF in promoting metastasis and tumor cell growth. (**A**) Knockdown of endogenous BAG2 by 2 siRNA oligos preferentially inhibited the migration ability of Saos2-R175H, Saos2-R248W and Saos2-R273H cells compared with Saos2-Con cells as determined by transwell assays. Left panel: representative images form a portion of the field. Right panel: quantification of average number and area of migrated cells/field. (**B**) Knockdown of endogenous BAG2 preferentially inhibited the migration ability of HCT116 p53$^{R248W/-}$ cells compared with HCT116 p53$^{-/-}$ cells. For **A**, **B**, date are presented as mean ±SD, n = 4. ***p < 0.001. (**C**) BAG2 knockdown greatly inhibited lung metastasis of HCT116 p53$^{R248W/-}$ cells but had a limited effect on HCT116 p53$^{-/-}$ cells in vivo. HCT116 p53$^{R248W/-}$ and HCT116 p53$^{-/-}$ cells stably infected with shRNA against BAG2 and their control cells were injected into the nude mice via the tail vein. The number and size of lung metastatic tumors were

*Figure 6. continued on next page*

*Figure 6. Continued*

determined at 6 weeks after inoculation. Left panel: representative H&E images of lung sections. Scale bar: 200 μm. Middle and Right panels: quantification of average number (middle panel) and area (right panel) of lung metastatic tumors, respectively. Date are presented as mean ±SD, n = 8/group. **p < 0.01; ***p < 0.001. (**D**) Knockdown of BAG2 by shRNA preferentially inhibited the anchorage-independent growth in HCT116p53$^{R248W/-}$ cells but not HCT116 p53$^{-/-}$ cells. Upper panel: representative images of cell colonies in soft agar. Lower panel: quantification of average number of colonies/field. Date are presented as mean ±SD, n = 4. ***p < 0.001. (**E**) BAG2 knockdown inhibited the growth of HCT116 xenograft tumors in a largely mutp53-dependent manner. HCT116 p53$^{R248W/-}$ and HCT116 p53$^{-/-}$ cells stably infected with shRNA against BAG2 and their control cells were employed for xenograft tumor formation in nude mice. Upper panel: A representative image of xenograft tumors. Lower panel: growth curves of xenograft tumors. Tumor volumes are presented as mean ±SD, n = 6/group. ***p < 0.001. (**F**) BAG2 knockdown decreased mutp53 protein levels in HCT116 p53$^{R248w/-}$ xenograft tumors as determined by Western blot assays.

The following figure supplements are available for figure 6:

**Figure supplement 1**. Knockdown of endogenous BAG2 by shRNA vectors inhibited mutp53 GOF in promoting migration in cells.

**Figure supplement 2**. BAG2 knockdown by shRNA inhibited the proliferation rate in HCT116 p53$^{R248W/-}$ cells but not HCT116 p53$^{-/-}$ cells.

important mechanism for BAG2 overexpression in tumors. We further investigated whether BAG2 overexpression is associated with poor prognosis in cancer patients by using the PrognoScan database. PrognoScan, which has a large collection of publicly available database with microarray data and clinical information, can assess the prognostic power of gene expression levels (*Mizuno et al., 2009*). As shown in *Figure 7B–E*, BAG2 overexpression is associated with poor disease free survival in colorectal cancer patients (HR = 1.40, p = 0.022), poor disease specific survival in lung cancer patients (HR = 2.4, p = 0.00001), poor relapse free survival in breast cancer patients (HR = 1.3, p = 0.00014) and poor distant recurrence free survival in soft tissue cancer patients (HR = 1.67, p = 0.00001). These results suggest the significant prognostic value of BAG2 expression levels for patients with various types of cancer.

The correlation between BAG2 overexpression and mutp53 accumulation was further investigated in a cohort of human colorectal cancer samples with known p53 mutation status and p53 protein levels (n = 100) (*Zheng et al., 2013*). p53 mutation status was determined by direct sequencing of exons 2–11 of p53 and the p53 protein levels were determined by IHC staining as previously described (*Zheng et al., 2013*). All tumors carrying mutp53 and a small percentage of tumors with wtp53 showed positive staining for p53 (>10% cells are stained). Tumors were divided into 2 groups according to median BAG2 expression levels as determined by Taqman real-time PCR assays. There is a clear correlation between high BAG2 expression and mutp53 accumulation (*Figure 7F*). In tumors with mutp53, 66.7% of tumors (18 out of 27) with high BAG2 expression displayed high p53 staining (>30% cells are stained) while only 40.9% tumors (9 out of 22) with low BAG2 expression had high p53 staining (p = 0.035). In contrast, in tumors with wtp53, there is no correlation between BAG2 expression and p53 accumulation. Among these tumors, 21.7% of tumors (5 out of 23) with high BAG2 expression and 17.9% of tumors (5 out of 28) with low BAG2 expression displayed low p53 staining (10–30% cells are stained), respectively (p = 0.36) (*Figure 7F*). These results demonstrate that BAG2 overexpression is significantly correlated with accumulation of mutp53 protein in colorectal cancers.

## Discussion

Many tumor-associated mutp53 proteins gain new oncogenic activities independently of wtp53, which is critical for mutp53 to promote tumorigenesis. While wtp53 proteins are kept at low levels in normal tissues under normal conditions, mutp53 proteins often accumulate to high levels in tumors, which is critical for mutp53 GOF in tumorigenesis (*Terzian et al., 2008*; *Oren and Rotter, 2010*; *Muller and Vousden, 2013*; *Liu et al., 2015*). Currently, the mechanism for mutp53 accumulation in tumors is poorly understood. Results from mouse genetic experiments with knockout of MDM2 in mutp53 mice

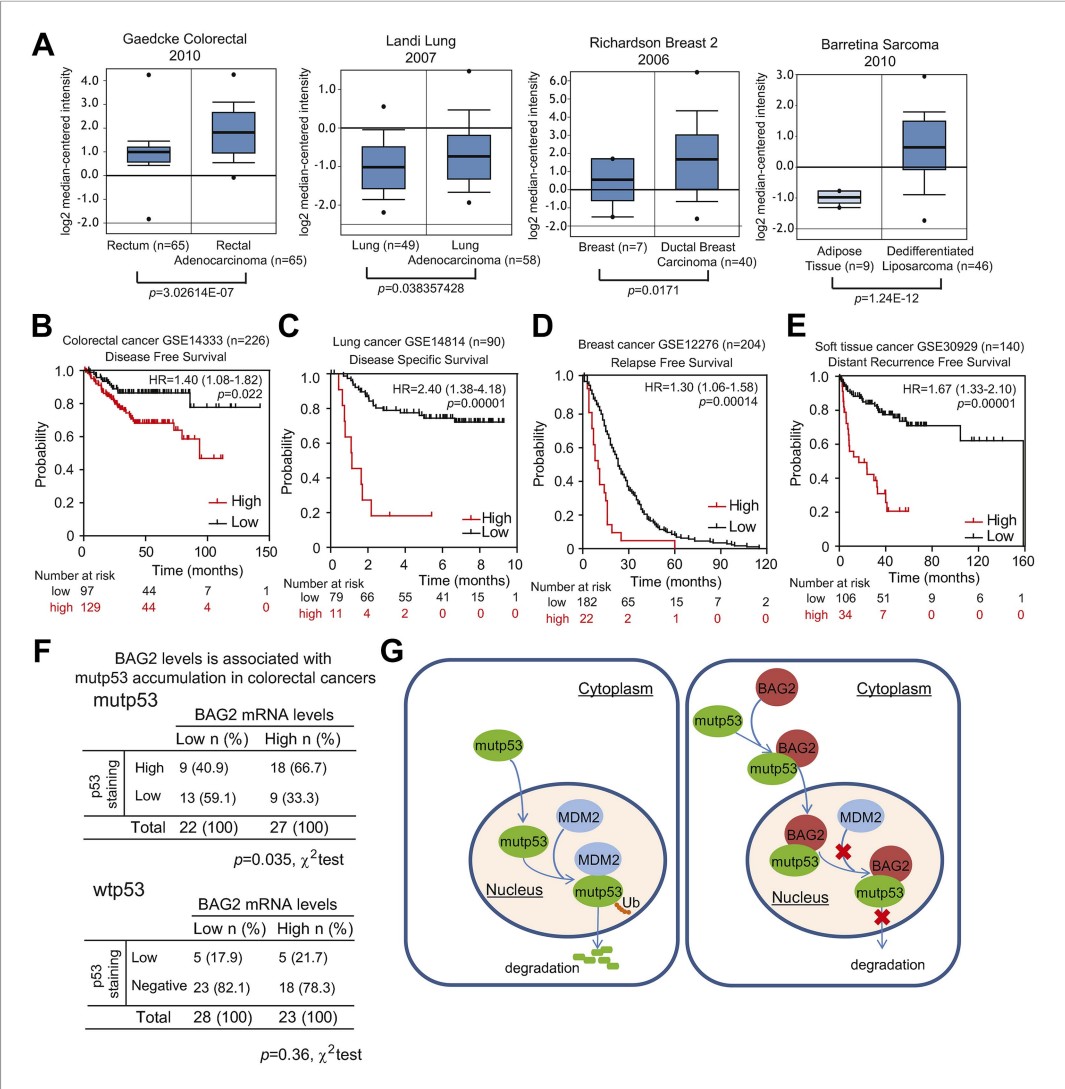

**Figure 7**. BAG2 is overexpressed in many human tumors and high levels of BAG2 are associated with mutp53 protein accumulation in human tumors. (**A**) BAG2 mRNA levels are elevated in human cancers, including colorectal cancers, lung cancers, breast cancers and sarcomas. BAG2 mRNA levels in normal and cancer tissues are presented as box plots based on data in four different datasets obtained from the Oncomine database. The expression levels of BAG2 are expressed in terms of a log2 median-centered intensity which is calculated by normalizing the intensity of BAG2 probe to the median of the probe intensities across the entire array. (**B**–**E**) High levels of BAG2 are associated with poor prognosis in cancer patients. Kaplan–Meier curves indicating the disease free survival of 226 colorectal cancer patients (**B**), the disease specific survival of 90 lung cancer patients (**C**), the relapse free survival of 204 breast cancer patients (**D**) and the distant recurrence free survival of 140 soft tissue cancer patients (**E**). The survival information and expression levels of BAG2 were obtained from the public available databases (GSE14333 for **B**, GSE14814 for **C**, GSE12276 for **D**, and GSE30929 for **E**) and analyzed by PrognoScan, a web based platform evaluating the prognostic power of gene expression levels. (**F**) BAG2 overexpression correlates with mutp53 protein accumulation (p = 0.036, $\chi 2$ test) but not wtp53 protein accumulation in human colorectal cancers. BAG2 mRNA levels were determined in human colorectal cancers and normalized with β-actin. (**G**) Schematic model depicting that mutp53 interacts with BAG2 and promotes BAG2 nuclear translocation to inhibit MDM2-mediated mutp53 protein degradation, which in turn promotes mutp53 protein accumulation and GOF in tumorigenesis.

The following figure supplement is available for figure 7:

**Figure supplement 1**. Amplification of the *BAG2* gene was observed in many human tumors.

and cell-based in vitro experiments suggest that MDM2 maintains mutp53 protein levels low in normal tissues, whereas some changes occurred in tumors disrupt MDM2-mediated mutp53 degradation, thereby leading to mutp53 accumulation. Recent studies have reported that HSP90 can bind to mutp53 and reduce mutp53 degradation mediated by MDM2. Knockdown of HSP90 by siRNA or blocking HDAC6-HSP90 axis by SAHA induced destabilization of mutp53 and inhibited its GOF in tumorigenesis (*Li et al., 2011a*, *2011b*). Our recent study showed that tumor-derived MDM2 short isoforms inhibited full-length MDM2-mediated mutp53 degradation, which promoted mutp53 accumulation and enhanced GOF in tumorigenesis (*Zheng et al., 2013*). In this study, we searched for the changes that occurred in tumors to disrupt MDM2-meidated mutp53 degradation by screening for mutp53 binding protein in tumors from mutp53 knock-in mice. Results from this study identified that BAG2 is a novel mutp53 binding protein that promotes mutp53 protein accumulation, which revealed a novel mechanism for mutp53 accumulation in tumor cells.

BAG2 belongs to the BAG family, which is characterized by the BAG domain. The BAG domain is a conserved region located at the C-terminus of the BAG-family proteins that binds to the ATPase domain of Hsc70 and has general nucleotide exchange activities towards Hsc70 (*Takayama and Reed, 2001*; *Kabbage and Dickman, 2008*). Therefore, proteins containing the BAG domain often functions as a co-chaperone protein. Aside from the formation of the BAG-Hsc70 interaction, BAG proteins functionally interact with many proteins to regulate cellular functions. The expression of BAG2 has been detected in many tissues. Recently, it was reported that BAG2 interacts with and stabilizes PINK1 and ataxin3-80Q, proteins involved in neurological diseases, through inhibiting their ubiquitination and degradation (*Che et al., 2013*, *2014*). BAG2 can also deliver Tau to the proteasome for protein degradation independently of ubiquitination (*Carrettiero et al., 2009*). In this study, we found that BAG2 preferentially binds to mutp53 at the DBD domain. BAG2 can interact with many different mutp53, including several tumor-associated mutational hotspots. This BAG2-mutp53 interaction is conserved in both human tumor cells and mouse tissues. It is unclear how BAG2 can discriminate between the DBD of a diverse range of mutp53 proteins and wtp53. BAG2 is a co-chaperone protein. It is possible that conformational changes of mutp53 proteins lead to its association with chaperone and co-chaperone proteins. It remains unclear whether BAG2 interacts with mutp53 directly or interacts with mutp53 through other protein, such as Hsc70, which will be of interest to investigate in future studies.

The role of BAG2 in tumor is poorly understood. In this study, we found that mutp53 interacted with BAG2 and promoted the translocation of BAG2 from the cytoplasm to the nucleus, where BAG2 inhibited the binding of MDM2 to mutp53 and the ubiquitination and degradation of mutp53 protein mediated by MDM2. It is unclear how the BAG2-mutp53 interaction interferes with the MDM2-mutp53 interaction since MDM2 binds to the N-terminus of mutp53 whereas BAG2 binds to the DBD of mutp53. Future studies are needed to further understand its mechanism. Results from this study showed that BAG2 promoted mutp53 protein accumulation in tumor cells, which in turn promoted mutp53 GOF in tumorigenesis (*Figure 7G*). Knockdown of endogenous BAG2 significantly inhibited cell proliferation, migration, metastasis and chemoresistance of tumor cells in a largely mutp53-dependent manner in cultured cells and/or in mice. These results strongly suggest that targeting BAG2 could be developed as a novel strategy to destabilize mutp53 and inhibit its GOF in tumorigenesis.

Importantly, analysis of several database from Oncomine showed that BAG2 is frequently overexpressed in many types of cancer (*Figure 7A*). Overexpression of BAG2 is significantly associated with poor prognosis in different types of cancer (*Figure 7B–E*). Furthermore, our results showed that BAG2 overexpression in colorectal tumors is significantly associated with mutp53 protein accumulation (*Figure 7F*). These results strongly suggest that BAG2 plays an important role in tumorigenesis and promoting mutp53 accumulation and GOF is a novel mechanism for BAG2 in tumorigenesis. It is unclear why not all tumors with BAG2 overexpression showed the accumulation of mutp53 protein. It is possible that additional mechanisms are involved in the regulation of mutp53 protein levels and/or the BAG2-mutp53 interaction. It will be interesting to examine whether BAG2 displays weak or no interaction with mutp53 protein in this subgroup of tumor samples in future studies. It is also worth noting that while normal tissues from Trp53$^{R172H/R172H}$ mice express a lot of BAG2, there is a limited amount of interacted BAG2-mutp53 protein complex and no clear accumulation of mutp53 proteins in the normal tissues (*Figure 1C* and *Figure 1—figure supplement 1*). These results suggest that some tumor-specific events might contribute to the effect of BAG2 on mutp53 accumulation.

Data from cBioportal showed amplification of the *BAG2* gene in many types of human tumors, suggesting that gene amplification is an important mechanism for BAG2 overexpression in human tumors. Considering that less than 10% of tumors had the amplification of BAG2 in majority of tumor types, it is possible that additional mechanisms contribute to BAG2 overexpression in tumors, which needs further investigation in future studies.

Taken together, results from this study demonstrate that BAG2 interacts with mutp53 to prevent its degradation by MDM2, leading to mutp53 accumulation in tumor cells and enhanced mutp53 GOF in tumorigenesis. Knockdown of BAG2 greatly reduces mutp53 protein levels in tumor cells and greatly compromises mutp53 GOF in tumorigenesis, including tumor growth, metastasis and chemoresistance. Considering that BAG2 is frequently overexpressed in cancer cells, our findings revealed a new and important mechanism for mutp53 protein accumulation in tumors. Trp53 is the most frequently-mutated gene in tumors. Mutp53 protein is frequently accumulated in tumors, which is critical for mutp53 GOF in tumor development. Therefore, mutp53 has become an extremely attractive target for tumor therapy. Our findings that BAG2 promotes mutp53 protein accumulation and mutp53 GOF in tumorigenesis strongly suggest that BAG2 could be a potential target for cancer therapy in tumors containing mutp53.

## Materials and methods

### Cell culture, mouse strains, constructs and cell treatments

Human lung cancer H1299, osteosarcoma Saos2, breast cancer SK-BR-3, MDA-MB-468, colorectal cancer HT29, SW480 and hepatocellular carcinoma Huh-7 cell lines were obtained from ATCC (Manassas, VA). Human HCT116 p53$^{R248W/-}$ cells were gifts from Dr Bert Vogelstein at Johns Hopkins University. Stable cell lines expressing mutp53 R175H, R248Q and R273H were established as previously described (*Zheng et al., 2013*). p53$^{-/-}$ mice were obtained from Jackson Laboratory (Bar Harbor, ME) and Trp53$^{R172H/R172H}$ mice were gifts from Dr Gigi Lozano at MD Anderson Cancer Center. Expression vectors of BAG2-HA (pcDNA-HA-BAG2) were gifts from Dr Cam Patterson at University of North Carolina. Expression vectors of mutp53 fragments containing different domains were obtained by using site-directed mutagenesis to introduce R175H mutation into expression vectors of wtp53 fragments containing different domains, which were generous gifts from Dr Xinbin Chen at University of California, Davis. R175H mutp53$^{NLS}$ expression vectors were obtained by using site-directed mutagenesis. Primers used for site mutagenesis and cloning for mutp53 fragments, Flag-tagged FL BAG2 and BAG2 fragments are listed in *Table 2*. Retroviral shRNA vectors against human BAG2 were purchased from Open Biosystems (Thermo Scientific, Waltham, MA, Cat#V2LHS-27769). Two different siRNA oligos against MDM2 were purchased from Qiagen (Germantown, MD, Cat#SI00300846) and Dharmacon (Lafayette, CO, Cat#M-003279-01). Two different siRNA oligos against BAG2 were purchased from IDT (Coralville, Iowa). siRNA targeting BAG2: siRNA-1: 5′-GUU GGC UUU AGC GUU GAU CUU CGC CUG-3′; siRNA-2: 5′-GUG UCA GUA GAA ACA AUU AGA AAC C-3′. 5-FU and MG132 were purchased from Sigma (St. Louis, MO).

### IP assays and IP coupled with LC-MS/MS assays

To determine mutp53 binding partners in mouse tissues, mouse mutp53 protein complexes were purified from lysates from tumor and normal tissues of mutp53$^{R172H/R172H}$ mice by IP using anti-p53 (FL393) beads and eluted with 0.1 M Glycine solution. Eluted materials were separated in a 4–15% Tris SDS gel and visualized by silver staining using the silver staining kit (Invitrogen, Grand Island, NY) and coomassie blue staining. Coomassie blue-stained protein bands were excised from the gel and subsequently analyzed by LC-MS/MS at the Biological MS facility of Rutgers University.

IP assays were performed as previously described (*Zheng et al., 2013*). In brief, 1 mg cell or tissue lysates in NP-40 buffer were used for IP using anti-p53 (DO-1 for human cells and FL393 for mouse tissues, Santa Cruz, Dallas, Texas), anti-HA and anti-Flag antibodies to pull down mutp53, BAG2-HA and BAG2-Flag protein, respectively.

### Western blot assays

Standard Western blot assays were used to analyze the levels of protein. Nuclear extracts were prepared by using Qproteome Nuclear Protein Kit (Qiagen). Antibodies against p53 (FL393; 1:2000 dilution; Santa Cruz), MDM2 (2A10; 1:1000 dilution), Flag (1:10,000 dilution; Sigma), BAG2 (1:1000

**Table 2.** Sequences of the primer sets used for site-directed mutagenesis and amplifying p53 and BAG2 fragments

| Name of fragments | | Primer sequences |
|---|---|---|
| **For site-directed mutagenesis** | | |
| Mutp53 R175H-HA P1 (aa 1–363), P2 (aa 43–393), P3 (aa 43–363) | Forward | 5′-GAG GTT GTG AGG CAC TGC CCC CAC CAT-3′ |
| | Reverse | 5′-ATG GTG GGG GCA GTG CCT CAC AAC CTC-3′ |
| R175H mutp53$^{NLS}$ | Forward 1 | 5′-GTT GGG CAG TGC TGC CGC AGT GCT CCC TGG GGG CAG-3′ |
| | Reverse 1 | 5′-CTG CCC CCA GGG AGC ACT GCG GCA GCA CTG CCC AAC-3′ |
| | Forward 2 | 5′-TGA AAT ATT CTC CAT CCA GTG GTG CCG CCG CTG GCT GGG GAG AGG AGC TGG TGT TGT TG-3′ |
| | Reverse2 | 5′-CAA CAA CAC CAG CTC CTC TCC CCA GCC AGC GGC GGC ACC ACT GGA TGG AGA ATA TTT CA-3′ |
| **For amplifying p53 and BAG2 fragments** | | |
| Mutp53 R175H-HA, P4 (aa 93–393) | Forward | 5′-GCG AAT TCA CCA TGG GCT ACC CAT ACG ATG TTC AGA TTA CGC TCT GTC ATC TTC TGT CCC TT-3′ |
| | Reverse | 5′-GAT CGA ATT CTC AGT CTG AGT CAG GCC CTT-3′ |
| Mutp53 R175H-HA, P5 (aa 93–325), wtp53-DBD, Mutp53 R248W-DBD Mutp53 R273H-DBD | Forward | 5′-GCG AAT TCA CCA TGG GCT ACC CAT ACG ATG TTC AGA TTA CGC TCT GTC ATC TTC TGT CCC TT-3′ |
| | Reverse | 5′-GCG AAT TCT CAT CCA TCC AGT GGT TTC TT-3′ |
| Mutp53 R175H-HA, P6 (Δaa 101–300) | Forward 1 | 5′-GCG AAT TCA CCA TGG GCT ACC CAT ACG ATG TTC AGA TTA CGC TGA GGA GCC GCA GTC AGA TC-3′ |
| | Reverse 1 | 5′-CTT AGT GCT CCC TGG CTG GAA GGA CAG A-3′ |
| | Forward 2 | 5′-TCT GTC CTT CCA GCC AGG GAG CAC TAA G-3′ |
| | Reverse 2 | 5′-GAT CGA ATT CTC AGT CTG AGT CAG GCC CTT-3′ |
| BAG2-Flag | Forward | 5′-CGG AAT TCA TGG CTC AGG CGA AGA-3′ |
| | Reverse | 5′-CGG GAT CCA TTG AAT CTG CTT TCA GCA T-3′ |
| BAG2 B1-Flag | Forward | 5′-CGG AAT TCA TGG CTC AGG CGA AGA-3′ |
| | Reverse | 5′-CGG GAT CCT CTT CCC ATC AAA CGG TT-3′ |
| BAG2 B2-Flag | Forward | 5′-CGG AAT TCA CCA TGG GAA GAA CTC TCA CCG TT-3′ |
| | Reverse | 5′-CGG GAT CCA TTG AAT CTG CTT TCA GCA T-3′ |

dilution; Aviva Systems Biology), HA (1:4000 dilution; Roche), α-Tubulin (C-5286; 1:1000 dilution; Santa Cruz), Lamin A/C (SC-7293; 1:1000 dilution; Santa Cruz), cleaved-caspase 3 (D175; 1:1000 dilution; Cell Signaling), and β-actin (1:20,000 dilution; Sigma) were used in this study.

## IF staining assays

IF staining was performed as previously described (*Zheng et al., 2013*). Antibodies against p53 (FL393) and Flag were used to detect p53 and BAG2-Flag, respectively. Slides were then incubated with Alexa Fluor 555 Goat Anti-Rabbit IgG (H + L) and Alexa Fluor 488 Goat Anti-mouse IgG (H + L) (Invitrogen). Nuclei were stained with 4′, 6-diamidino-2-phenylindole (DAPI; Vector, Burlingame, CA).

## Quantitative real-time PCR

RNA from cells was prepared with RNeasy kit (Qiagen). RNA from formalin fixed and paraffin-embedded colorectal tumor sections was prepared with High Pure miRNA Isolation kit (Roche, Indianapolis, IN). The cDNA was prepared by using High Capacity cDNA Reverse Transcription Kit (Applied Biosystems, Grand Island, NY). Primers for Taqman real-time PCR assays were purchased from Applied Biosystems. The expression of genes was normalized with the β-actin gene.

## In vivo ubiquitination of Mutp53

In vivo ubiquitination assays were performed as previously described (*Liu et al., 2014*). In brief, cells were transfected with different expression vectors, including mutp53 R175H, BAG2-HA and His-ubiquitin, or transfected with siRNA against BAG2 together with His-ubiquitin expression vectors. At 24 hr after transfection, cells were treated with MG132 for 6 hr. The levels of mutp53 ubiquitination were determined by IP using DO-1 antibody followed by Western blot assays with an anti-ubiquitin antibody (P4D1; 1:1000; Santa Cruz).

## Annexin V staining

Annexin V staining was used to determine apoptosis as previously described (*Yu et al., 2014*). In brief, cells were stained by using Muse Annexin V and Dead Cell Assay Kit (Millipore) and analyzed in a bench flow cytometry, the Muse Cell Analyzer (Millipore, Billerica, MA).

## Cell migration assays

The transwell system (BD Biosciences, San Jose, CA) was employed for cell migration assays as previously described (*Zheng et al., 2013*). In brief, cells in FBS-free medium were seeded into upper chambers. The lower chamber was filled with medium supplemented with 10% FBS. Cells on the lower surface of upper chambers were counted after culturing at 37°C for 24 hr.

## Anchorage-independent growth assays

Anchorage-independent growth assays were performed as previously described (*Li et al., 2014*). In brief, cells were seeded in 6-well plates coated with media containing 0.6% agarose, and cultured in media containing 0.3% agarose. Colonies were stained and counted after 2–3 weeks.

## Xenograft tumorigenicity assays

Cells ($5 \times 10^6$ in 0.2 ml PBS) were injected subcutaneously (s.c.) into 8-week-old BALB/c athymic nude mice (Taconic). Tumor volumes were measured every 2 days for 3 weeks. Tumor volume = 1/2 (length $\times$ width$^2$) (n = 6 mice/group). Tumor samples were processed for routine histopathological examination.

## In vivo metastasis assays

In vivo lung metastasis assays were performed as previously described (*Zheng et al., 2013*). In brief, HCT116 p53$^{R248W/-}$ and HCT116 p53$^{-/-}$ cells with or without knockdown of BAG2 by shRNA vectors ($1 \times 10^6$ in 0.1 ml PBS) were injected into 8-week-old nude mice via the tail vein (n = 8 mice/group). The mice were sacrificed at 6 weeks after the inoculation. The numbers of lung tumors were counted under a dissecting microscope and confirmed by histopathological analysis. The areas of tumor nodules were quantified in 8 representative images taken at 10 × magnification by using the imageJ software. Animal protocols were approved by the IACUC committee of Rutgers University.

## Database of cancer patients

PrognoScan (http://www.prognoscan.org/), which has a large collection of publicly available database with microarray data and clinical information (*Mizuno et al., 2009*), was used to analyze the prognostic

power of BAG2 expression levels in colorectal cancer patients (GSE14333, *Sieber et al., 2010*), lung cancer patients (GSE14814, *Tsao et al., 2010*), breast cancer patients (GSE12276, *Bos et al., 2009*), and soft tissue cancer patients (GSE30929, *Gobble et al., 2011*).

A cohort of the de-identified colorectal cancer tissues with known p53 mutation status and p53 protein levels was obtained from the database of the First Affiliated Hospital of Harbin Medical University (Harbin, China) with an IRB approval (*Zheng et al., 2013*). None of these patients received pre-surgical chemotherapy.

## Statistical analysis

The differences in xenograft tumor growth among groups were analyzed for statistical significance by ANOVA, followed by Student's *t*-tests using a GraphPad Prism software. Kaplan–Meier statistics were performed to analyze the significance of differences in survival of patients among different groups using software in PrognoScan website. All other p values were obtained using Student's *t*-test or $\chi 2$ test. Values of $p < 0.05$ were considered to be significant.

## Acknowledgements

We thank Dr Arnold Levine for helpful discussion. WH is supported by the grants from NIH (1R01CA160558), the Ellison Foundation, the New Jersey Health foundation, and the New Investigator Award of CINJ. ZF is supported by the grants from NIH (1R01CA143204) and Busch Biomedical Grant.

## Additional information

### Funding

| Funder | Grant reference | Author |
|---|---|---|
| National Cancer Institute (NCI) | 1R01CA160558 | Wenwei Hu |
| National Cancer Institute (NCI) | 1R01CA143204 | Zhaohui Feng |

The funder had no role in study design, data collection and interpretation, or the decision to submit the work for publication.

### Author contributions

XY, Conception and design, Acquisition of data, Analysis and interpretation of data, Drafting or revising the article; YZ, Conception and design, Acquisition of data, Analysis and interpretation of data; JL, CZ, HY, JW, TZ, LL, JL, Acquisition of data, Analysis and interpretation of data; ZF, WH, Conception and design, Analysis and interpretation of data, Drafting or revising the article

### Ethics

Animal experimentation: This study was performed in strict accordance with the recommendations in the Guide for the Care and Use of Laboratory Animals of the National Institutes of Health. All of the animals were handled according to approved institutional animal care and use committee (IACUC) protocol (#I13-028) of Rutgers University.

## Additional files

### Major datasets

The following previously published datasets were used:

| Author(s) | Year | Dataset title | Dataset ID and/or URL | Database, license, and accessibility information |
|---|---|---|---|---|
| Gaedcke J, Grade M, Jung K, Camps J, Jo P, Emons G, Gehoff A, Sax U, Schirmer M, Becker H, Beissbarth T, Ried T, Ghadimi M | 2011 | Mutated KRAS induces overexpression of DUSP4, a MAP-kinase phosphatase, and SMYD3, a histone methyltransferase, in rectal carcinomas | http://www.ncbi.nlm.nih.gov/geo/query/acc.cgi?acc=GSE20842 | Publicly available at the NCBI Gene Expression Omnibus (Accession on. GSE20842). |

| Author(s) | Year | Dataset title | Dataset ID and/or URL | Database, license, and accessibility information |
|---|---|---|---|---|
| Landi M, Dracheva T, Rotunno M, Figueroa JD, Liu H, Dasgupta A, Mann FE, Fukuoka J, Hames M, Bergen AW, Murphy SE, Yang P, Pesatori AC, Consonni D, Bertazzi P, Wacholder S, Shih JH, Caporaso NE, Jen J | 2008 | Gene expression signature of cigarette smoking and its role in lung adenocarcinoma development and survival | http://www.ncbi.nlm.nih.gov/geo/query/acc.cgi?acc=GSE10072 | Publicly available at the NCBI Gene Expression Omnibus (Accession on. GSE10072). |
| Richardson A | 2006 | Human breast tumor expression | http://www.ncbi.nlm.nih.gov/geo/query/acc.cgi?acc=GSE3744 | Publicly available at the NCBI Gene Expression Omnibus (Accession on. GSE3744). |
| Taylor BS, Barretina J, Meyerson M, Singer S | 2010 | Whole-transcript expression data for soft-tissue sarcoma tumors and control normal fat specimens | http://www.ncbi.nlm.nih.gov/geo/query/acc.cgi?acc=GSE21122 | Publicly available at the NCBI Gene Expression Omnibus (Accession on. GSE21122). |
| Sieber OM | 2010 | Expression data from 290 primary colorectal cancers | http://www.ncbi.nlm.nih.gov/geo/query/acc.cgi?acc=GSE14333 | Publicly available at the NCBI Gene Expression Omnibus (Accession on. GSE14333). |
| Tsao MS, Zhu CQ, Ding K, Strumpt D | 2010 | Prognostic and Predictive Gene Signature for Adjuvant Chemotherapy in Resected Non-Small-Cell Lung Cancer | http://www.ncbi.nlm.nih.gov/geo/query/acc.cgi?acc=GSE14814 | Publicly available at the NCBI Gene Expression Omnibus (Accession on. GSE14814). |
| Bos PD, Zhang XH, Nadal C, Shu W, Gomis RR, Nguyen DX, Minn AJ, van de Vijver MJ, Gerald WL, Foekens JA, Massagué J | 2009 | Expression data from primary breast tumors | http://www.ncbi.nlm.nih.gov/geo/query/acc.cgi?acc=GSE12276 | Publicly available at the NCBI Gene Expression Omnibus (Accession on. GSE12276). |
| Gobble RM, Qin LX, Brill ER, Angeles CV, Ugras S, O'Connor RB, Moraco NH, Decarolis PL, Antonescu C, Singer S | 2011 | Whole-transcript expression data for liposarcoma | http://www.ncbi.nlm.nih.gov/geo/query/acc.cgi?acc=GSE30929 | Publicly available at the NCBI Gene Expression Omnibus (Accession on. GSE30929). |

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
