## [Decision Letter]

Thank you for submitting your work entitled “BAG2 Promotes Tumorigenesis through Enhancing Mutant p53 Protein Levels and Function” for peer review at *eLife*. Your submission has been favorably evaluated by James Manley (Senior Editor), a Reviewing Editor, and three reviewers, whose detailed comments are included below.

The reviewers have discussed the reviews with one another and the Reviewing Editor has drafted this decision to help you prepare a revised submission.

Essential revisions:

Despite overall positive reviews on your paper, there were two issues that were raised during consultation between the reviewers and the member of the board that is handling this paper.

First, with the exception of Figure 2—figure supplement 1 where the authors do a co-IP in SK-BR-3 cells, all other experiments in human cells are in either H1299 and Saos2 p53 null cells or HCT116 cells, each of which came from tumors that did not have mutant p53. It is felt that the authors should examine more cancer cell lines with endogenously expressed mutant p53 to determine the generality of their findings that BAG2 and mutant p53 interact.

Second, the authors should examine normal tissues (or cells therefrom) expressing mutant p53 where there is a lot of BAG2 but no stabilization of p53 to see if in this setting BAG2 binds to mutant p53. This might provide insight into Reviewer 3's cogent comments.

Minor points:

1) The authors are encouraged to test whether BAG2 binds to p53 directly in an in vitro binding assay, or to state clearly in the Discussion that whether BAG2 binds to p53 directly, or is brought by Hsc70, is not yet clear.

2) In the subsection “BAG2 Is A Novel Mutp53-interacting Protein in p53^R172H/R172H^ Mouse Tumors and Human Cells”, please tell us how many thymic lymphomas were used for the IP/LC-MS/MS experiments.

3) In the subsection “BAG2 Promotes Mutp53 GOF in Chemoresistance”, the authors indicate that BAG2 greatly increased 5-FU induced apoptosis. It was only 50%, so this statement needs to be toned down a bit.

4) Figure 7 needs to be described a little better in the legend. I do not know what the horizontal line at 0 stands for. Shouldn't the normal controls be at zero?

5) The Discussion was simply a retelling of the experiments performed and not really a discussion. The authors need to discuss their findings. Why, for example, do some tumors have increased BAG but no accumulation of mutant p53.

Reviewer #1:

In this manuscript, Yue and colleagues make effort to solve the conundrum in the p53 field about what controls the levels of mutant p53 in tumor cells. Previously, it was believed that because mutant p53 does not transactivate its ubiquitin ligase MDM2, the level of mutant p53 in cells is very high. However, two groups created mutant p53 knock-in mice, and failed to see high levels of mutant p53 in tissues from these mice, only in tumors, thus dispelling this hypothesis. Yue and colleagues solve this issue, and this is a very significant finding, not just for the p53 field, but for the field of cancer, as mutant p53 is known to contribute to tumor aggressiveness in many ways.

The authors perform mass spec of mutant-p53 interacting proteins in normal and tumor cells and (in addition to reporting the entire list of mutant p53 binding proteins, which is important) they show that BAG2 is a major binder. They prove the interaction occurs for endogenous mutant p53 and BAG2, they map the interaction domains to the DNA binding domain of p53 and the BAG domain of BAG2. Importantly, they show that BAG2 is translocated to the nucleus by mutant, but not wtp53, and that silencing BAG2 destabilizes mutant p53 by removing the ability of BAG2 to inhibit interaction between mutant p53 and MDM2. In an elegant series of experiments, the authors show that silencing BAG2 inhibits the “gain of function” activity of mutant p53 (resistance to 5FU, migration and metastasis), and further they show that silencing BAG2 inhibits the development of tumors with mutant p53, but not null for p53. The authors go on to show that elevated BAG2 is a marker of poor prognosis in several tumor types, and that BAG2 mRNA levels are associated with mutant p53 staining.

Overall, the manuscript is well-controlled, well-written and highly significant.

Reviewer #2:

In this study, the authors identify a series of proteins from tumors in mice that bind p53^R172H^ and characterize BAG2. They characterize the domains of each protein that interact and show that BAG2 prevents mutant p53 interactions with MDM2. BAG2 knockdown experiments showed decreased migration in tissue culture and decreased metastasis in nude mice. The last figure shows correlations in human cancers.

Reviewer #3:

This is an interesting paper and the data presented are exceptionally clear and convincing. I don't have any criticisms of the work as it stands. My questions are more around the mechanism. Why has BAG2 evolved to specifically bind mutant p53 and how can it discriminate between the DNA binding domains of a diverse range of mutant p53 s compared to wild type? How does BAG2 interfere with MDM2 binding? I don't think the authors can answer all of these questions but they might pose them and speculate.

Interestingly, Figure 1 shows that in the normal tissue of mutant p53 expressing mice, there seems to be no effect of BAG2 on mp53 stabilization, despite lots of BAG2 expression. It might be useful to explore this a bit more – is the BAG2 not bound to p53 here? Why not? There must be some tumor-specific event that allows BAG2 to stabilize p53, but this is independent of BAG2 expression levels (in this case I'm not sure what the relevance of overexpression of BAG2 in tumors really is). Maybe this could be explored in normal and transformed versions of the same cell line?

---

## [Author Response]

Despite overall positive reviews on your paper, there were two issues that were raised during consultation between the reviewers and the member of the board that is handling this paper.

*First, with the exception of*
Figure 2—figure supplement 1
*where the authors do a co-IP in SK-BR-3 cells, all other experiments in human cells are in either H1299 and Saos2 p53 null cells or HCT116 cells each of which came from tumors that did not have mutant p53. It is felt that the authors should examine more cancer cell lines with endogenously expressed mutant p53 to determine the generality of their findings that BAG2 and mutant p53 interact.*

Thanks for this very good suggestion. We agree with the reviewers that additional data obtained from more cancer cell lines with endogenous mutp53 can provide important information about the generality of BAG2-mutp53 interaction. We have employed human colorectal cancer cell lines HT-29 (mutp53 R273H), SW480 (mutp53 R273H), human breast cancer cell line MDA-MB-468 (mutp53 R273H) and human hepatocellular cancer cell line Huh-7 (mutp53 Y220C) to examine the interaction between the endogenous BAG2 and mutp53 proteins by co-IP assays. In the revised manuscript, we have presented new data showing that the BAG2-mutp53 interaction was observed in all these cell lines. These data in revised Figure 2—figure supplement 1 strongly suggest the generality of BAG2-mutp53 interaction in human tumor cells.

Second, the authors should examine normal tissues (or cells therefrom) expressing mutant p53 where there is a lot of BAG2 but no stabilization of p53 to see if in this setting BAG2 binds to mutant p53. This might provide insight into reviewer#3's cogent comments.

This is a very good suggestion. The reviewers raised a very good point that while normal thymus tissues from mutp53^R172H/R172H^ mice express a lot of BAG2, there is no apparent effect of BAG2 on mutp53 accumulation in normal tissues (Figure 1) (see detailed response to Reviewer 3). The reviewers provided a very good suggestion to carefully analyze the BAG2-mutp53 interaction in normal tissues from mutp53^R172H/R172H^ expressing mice. We re-analyzed the raw data for Figure 1. We observed lower levels of mutp53 protein and weak interaction between BAG2 and mutp53 in normal thymus tissues from mutp53^R172H/R172H^ expressing mice after longer exposure. We have presented the film with longer exposure time in revised Figure 1. In addition, we examined the interaction between BAG2 and mutp53 in normal kidney and spleen tissues in addition to thymus tissues collected from mutp53^R172H/R172H^ expressing mice. In all these tissues, we observed weak interaction between BAG2 and mutp53. These data are presented in Figure 1—figure supplement 1, which suggest that some additional tumor-specific changes might contribute to the effect of BAG2 on mutp53 accumulation. We have added the following to the Discussion: “It is also worth noting that while normal tissues from p53^R172H/R172H^ mice express a lot of BAG2, there is a limited amount of interacted BAG2-mutp53 protein complex and no clear accumulation of mutp53 proteins in the normal tissues (Figure 1 and Figure 1—figure supplement 1). These results suggest that some tumor-specific events might contribute to the effect of BAG2 on mutp53 accumulation.”

Minor points:

*1) The authors are encouraged to test whether BAG2 binds to p53 directly in an* in vitro *binding assay, or to state clearly in the Discussion that whether BAG2 binds to p53 directly, or is brought by Hsc70, is not yet clear.*

Thanks for this very good suggestion. It is unclear whether BAG2 binds to mutp53 directly or indirectly. As suggested by the reviewer, we have stated this in the Discussion: “It remains unclear whether BAG2 interacts with mutp53 directly or interacts with mutp53 through other protein, such as Hsc70, which will be of interest to investigate in future studies.”

*2) In the subsection “BAG2 Is A Novel Mutp53-interacting Protein in p53*^*R172H/R172H*^
*Mouse Tumors and Human Cells”, please tell us how many thymic lymphomas were used for the IP/LC-MS/MS experiments.*

We have added the number (n = 3) of thymic lymphomas used for the IP/LC-MS/MS assays in the subsection “BAG2 Is A Novel Mutp53-interacting Protein in Trp53^R172H/R172H^ Mouse Tumors and Human Cells”.

3) In the subsection “BAG2 Promotes Mutp53 GOF in Chemoresistance”, the authors indicate that BAG2 greatly increased 5-FU induced apoptosis. It was only 50%, so this statement needs to be toned down a bit.

As the reviewer suggested, we have changed the statement to: “knockdown of BAG2 increased 5-FU-induced apoptosis in Saos2-R175H…”.

*4)*
Figure 7
*needs to be described a little better in the legend. I do not know what the horizontal line at 0 stands for. Shouldn't the normal controls be at zero?*

This is a very good suggestion. We have provided the explanation for the calculation of the intensity of BAG2 expression presented in Figure 7 in the corresponding legend. The BAG2 expression levels for the normal controls may have a value less than zero when BAG2 is expressed at a level less than the median and may have a value greater than zero when BAG2 is expressed at a level higher than the median.

5) The Discussion was simply a retelling of the experiments performed and not really a discussion. The authors need to discuss their findings. Why, for example, do some tumors have increased BAG but no accumulation of mutant p53.

This is a very good suggestion. We further discussed our findings in the Discussion. In particular, we stated that not all tumors with BAG2 overexpression showed the accumulation of mutp53 protein: “It is unclear why not all tumors with BAG2 overexpression showed the accumulation of mutp53 protein […]. It will be interesting to examine whether BAG2 has weak or no interaction with mutp53 protein in this subgroup of tumor samples in future studies.”

Reviewer #3:

*This is an interesting paper and the data presented are exceptionally clear and convincing. I don't have any criticisms of the work as it stands. My questions are more around the mechanism. Why has BAG2 evolved to specifically bind mutant p53 and how* can *it discriminate between the DNA binding domains of a diverse range of mutant p53 s compared to wild type? How does BAG2 interfere with MDM2 binding? I don't think the authors* can *answer all of these questions but they might pose them and speculate.*

These are very good questions. While currently we don’t have final answers for these questions, we added following to the Discussion: “In this study, we found that BAG2 preferentially binds to mutp53 at the DBD domain […]. It remains unclear whether BAG2 interacts with mutp53 directly or interacts with mutp53 through other protein, such as Hsc70, which will be of interest to investigate in future studies.” Also: “It is unclear how the BAG2-mutp53 interaction interferes with the MDM2-mutp53 interaction since MDM2 binds to the N- terminus of mutp53 whereas BAG2 binds to the DBD of mutp53. Future studies are needed to further understand its mechanism.”